



# Iron from coal combustion particles dissolves much faster than mineral dust under simulated atmospheric acid conditions

Clarissa Baldo[1], Akinori Ito[2], Michael D. Krom[3,4], Weijun Li[5], Tim Jones[6], Nick Drake[7], Konstantin Ignatyev[8], Nicholas Davidson[1], Zongbo Shi[1]

[1]School of Geography Earth and Environmental Sciences, University of Birmingham, Birmingham, United Kingdom

[2]Yokohama Institute for Earth Sciences, JAMSTEC, Yokohama, Kanagawa 236-0001, Japan

[3]Morris Kahn Marine Station, Charney School of Marine Sciences, University of Haifa, Haifa, Israel

[4]School of Earth and Environment, University of Leeds, Leeds, United Kingdom

[5]Department of Atmospheric Sciences, School of Earth Sciences, Zhejiang University, Hangzhou 310027, China

[6]School of Earth and Environmental Sciences, Cardiff University, Cardiff, United Kingdom

[7]Department of Geography, King's College London, London, United Kingdom

[8]Diamond Light Source, Didcot, Oxfordshire, United Kingdom

*Correspondence to*: Zongbo Shi (z.shi@bham.ac.uk); Akinori Ito (akinori@jamstec.go.jp)

**Abstract.** Mineral dust is the largest source of aerosol iron (Fe) to the offshore global ocean, but acidic processing of coal fly ash (CFA) in the atmosphere may result in a disproportionally higher contribution of bioavailable Fe. Here, we determined the Fe speciation and dissolution kinetics of CFA from Aberthaw (United Kingdom), Krakow (Poland), and Shandong (China) in solutions which simulate atmospheric acidic processing. In CFA-$PM_{10}$ fractions, 8%-21.5% of the total Fe was as hematite and goethite (dithionite extracted Fe), 2%-6.5 % as amorphous Fe (ascorbate extracted Fe), while magnetite (oxalate extracted Fe) varied from 3%-22%. The remaining 50%-87 % of Fe was associated with aluminosilicates. High concentration of ammonium sulphate (($NH_4$)$_2SO_4$), often found in wet aerosols, increased Fe solubility of CFA up to 7 times at low pH (2-3). Our results showed a large variability in the effects of oxalate on the Fe dissolution rates at pH 2, from no impact in Shandong ash to doubled dissolution in Krakow ash. However, this enhancement was suppressed in the presence of high concentration of ($NH_4$)$_2SO_4$. Dissolution of highly reactive Fe was insufficient to explain the high Fe solubility at low pH in CFA, and the modelled dissolution kinetics suggests that other Fe phases such as magnetite may also dissolve rapidly under acidic conditions. Overall, Fe in CFA dissolved up to 7 times faster than in Saharan dust samples at pH 2. Based on these laboratory data, we developed a new scheme for the proton- and oxalate- promoted Fe dissolution of CFA, which was implemented into the global atmospheric chemical transport model IMPACT. The revised model showed a better agreement with observations of surface concentration of dissolved Fe in aerosol particles over the Bay of Bengal, due to the rapid Fe release at the initial stage at highly acidic conditions. The improved model also enabled us to predict sensitivity to a more dynamic range of pH changes, particularly between anthropogenic combustion and biomass burning aerosols.



## 1 Introduction

The availability of iron (Fe) limits primary productivity in high-nutrient low-chlorophyll (HNLC) regions of the global ocean including the subarctic North Pacific, the East Equatorial Pacific and the Southern Ocean (Boyd et al., 2007; Martin, 1990). In other regions of the global ocean such as the subtropical North Atlantic, the Fe input may affect primary productivity by stimulating nitrogen fixation (Mills et al., 2004; Moore et al., 2006). These areas are particularly sensitive to changes in the supply of bioavailable Fe. Atmospheric aerosols are an important source of soluble (and, thus potentially bioavailable) Fe to the offshore global ocean. The deposition of bioavailable Fe to the ocean can alter biogeochemical cycles and increase the carbon uptake, consequently affecting the climate (e.g., Jickells and Moore, 2015; Jickells et al., 2005; Kanakidou et al., 2018; Mahowald et al., 2010; Shi et al., 2012). In general, bioavailable Fe consists of aerosol dissolved Fe, and Fe-nanoparticles which can be present in the original particulate matter and/or formed during atmospheric transport as a result of cycling into and out of clouds (Shi et al., 2009). It is in addition possible that other more refractory forms of Fe could be solubilised in the surface waters by zooplankton (Schlosser et al., 2018) or the microbial community (Rubin et al., 2011).

Atmospheric Fe is largely derived from lithogenic sources, which contribute around 95% of the total Fe in suspended particles (e.g., Myriokefalitakis et al., 2018) and hence most studies concentrate on atmospheric processing of mineral dust (e.g., Cwiertny et al., 2008; Fu et al., 2010; Ito and Shi, 2016; Shi et al., 2011a; Shi et al., 2015). Mineral dust has low Fe solubility (dissolved Fe/ total Fe) near the source regions, generally below 0.5% (e.g., Schroth et al., 2009; Shi et al., 2011c), increasing somewhat as a result of atmospheric processing (e.g., Baker et al., 2021; Baker et al., 2020). Other sources of bioavailable Fe to the ocean are from combustion sources such as biomass burning, coal combustion and oil combustion (e.g., shipping emissions) (e.g., Ito et al., 2018; Rathod et al., 2020). Although these sources are only a small fraction of the total Fe in atmospheric particulates, the Fe solubility of pyrogenic sources can be 1–2 orders of magnitude higher than in mineral dust, and thus can be important in promoting carbon uptake. However the Fe solubility of these sources vary considerably depending on the particular sources with higher values observed for oil combustion and biomass burning (Ito et al., 2021b and references therein).

Wang et al. (2015) estimated that coal combustion produces around ~0.9 Tg yr$^{-1}$ of atmospheric Fe (on average for 1960–2007), contributing up to ~86% of the total anthropogenic Fe emissions. A more recent study, which has included metal smelting as atmospheric Fe source, estimated that coal combustion emitted ~0.7 Tg yr$^{-1}$ of Fe for the year 2010, contributing around 34% of the total anthropogenic Fe (Rathod et al., 2020). Although the use of coal as a principle energy source has been recently reduced as a result of concern about air quality and global warming, coal is still an important energy source in a number of countries in particular in the Asia-Pacific region (BP, 2020). In China, most of the total energy is supplied by coal, contributing over 50% of the global coal consumption in 2019, followed by India (12%), and the US (8%). Germany and Poland are the largest coal consumers in Europe, accounting together for around 40% of the European usage (BP, 2020). South Africa is also among the principal countries for coal consumption (BP, 2020) and is a source of particles to the Fe-limited Southern Ocean (e.g., Ito et al., 2019).

Coal fly ash (CFA) is a by-product of coal combustion. This generally consists of glassy spherical particles (e.g., Brown et al., 2011), which are formed through different transformations (decomposition, fusion, agglomeration, volatilization) of mineral matter in coal during combustion (e.g., Jones, 1995), and are transported with the flue gases undergoing rapid solidification. CFA are co-emitted with acidic gases such as sulphur dioxide ($SO_2$), nitrogen oxides ($NO_x$) and carbon dioxide ($CO_2$) (e.g., Munawer, 2018).

During long-range transport, CFA particles undergo atmospheric processing with the CFA surface coated by acidic species such as sulphuric acid ($H_2SO_4$) and oxalic acid ($H_2C_2O_4$) in atmospheric aerosols. Aged CFA particles are hygroscopic and

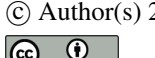



absorb water at typical relative humidity in the marine atmosphere. This forms a thin layer of water with high acidity, low pH
and high ionic strength (Meskhidze et al., 2003; Spokes and Jickells, 1995; Zhu et al., 1992). In addition, ammonia ($NH_3$)
which is a highly hydrophilic gas, can also partition into the aerosol phase, react with $H_2SO_4$ and form ammonium sulphate
(($NH_4)_2SO_4$) an important inorganic salt contributing to the high ionic strength in such atmospheric aerosols (Seinfeld and
Pandis, 2016).
At low pH conditions, Fe solubility in aerosols increases, as the high concentration of protons ($H^+$) weakens the Fe-O bonds
facilitating the detachment of Fe from the surface lattice (Furrer and Stumm, 1986). Li et al. (2017) provided the first
observational evidence to confirm that the acidification leads to the release of Fe from anthropogenic particles.
In addition to these inorganic processes, organic ligands can also enhance atmospheric Fe dissolution by forming soluble
complexes with Fe (e.g., Cornell and Schwertmann, 2003). For example, $H_2C_2O_4$ is an important organic species in
atmospheric aerosols (e.g., Kawamura and Bikkina, 2016). Laboratory studies have demonstrated that $H_2C_2O_4$ increases Fe
solubility of atmospheric aerosol sources (Chen and Grassian, 2013; Paris and Desboeufs, 2013; Paris et al., 2011; Xu and
Gao, 2008). Recently, observations over the Bay of Bengal indicate that $H_2C_2O_4$ contributes to the increase in atmospheric
water dissolved Fe (Bikkina et al., 2020).
To simulate the Fe dissolution in CFA, it is necessary to determine the dissolution kinetics under realistic conditions. Previous
studies have investigated the Fe dissolution kinetics of CFA under acidic conditions. Chen et al. (2012) simulated acidic and
cloud processing of certified CFA. Fu et al. (2012) determined the dissolution kinetics of CFA samples at pH 2, while Chen
and Grassian (2013) investigated the effect of organic species (e.g., oxalate and acetate) at pH 2-3. These studies showed that
high acidity and the presence of oxalate enhanced Fe dissolution, similar to those reported in mineral dust (Chen et al., 2012;
Chen and Grassian, 2013; Fu et al., 2012; Ito and Shi, 2016; Shi et al., 2011a). They also demonstrated that there are large
differences in dissolution rates in different types of CFA, likely related to Fe speciation.
Furthermore, high ionic strength, commonly seen in aerosol water, affects the activity of molecular species present in solution,
consequently it can significantly impact the Fe dissolution behaviour. Recent studies have considered the effect of the high
ionic strength on the Fe dissolution kinetic of CFA under acidic conditions. For example, the Fe solubility of CFA samples
was measured at pH 1-2 with high sodium chloride (NaCl) concentrations (Borgatta et al., 2016), and with high sodium nitrate
($NaNO_3$) concentrations Kim et al. (2020). In real atmospheric conditions, NaCl or $NaNO_3$ are unlikely to be the main driver
of high ionic strength in aged CFA. Although NaCl can coagulate with dust particles in the marine boundary layer (Zhang et
al., 2003), the aging of coal fly ash is primarily by the uptake of secondary species, particularly sulphate and ammonia (Li et
al., 2003). Ito and Shi (2016) found that at low pH and high concentration of $(NH_4)_2SO_4$ the Fe solubility of mineral dust is
likely to be enhanced by the adsorption of sulphate ions on the particle surface. However, to date the effect of high $(NH_4)_2SO_4$
concentrations on the Fe dissolution behaviour in combustion sources in the presence or absence of oxalate remains unknow.
The dissolution kinetics measured by Chen and Grassian (2013) has been used to develop a modelled dissolution scheme for
CFA, assuming a single Fe phase in CFA (Ito, 2015). However, there are multiple Fe phases in CFA, primarily hematite,
magnetite and Fe in aluminium silicate glass (Brown et al., 2011; Chen et al., 2012; Fu et al., 2012; Kukier et al., 2003; Kutchko
and Kim, 2006; Lawson et al., 2020; Sutto, 2018; Valeev et al., 2019; Waanders et al., 2003; Wang, 2014; Zhao et al., 2006),
but also accessory Fe-bearing minerals for example silicates, carbonate, sulphides and sulphates (Zhao et al., 2006). These
phases have a range of reactivities. Previous studies showed that CFA dissolves much faster during the first 1-2 hours than
subsequently (Borgatta et al., 2016; Chen et al., 2012; Chen and Grassian, 2013; Fu et al., 2012; Kim et al., 2020), confirming
the large difference in Fe dissolution from different phases.



In this study, laboratory experiments were conducted to determine the dissolution kinetics of coal combustion sources (e.g.,
coal fly ash) during simulated atmospheric acidic processing in the presence of $(NH_4)_2SO_4$ and oxalate which are commonly
found in atmospheric aerosols. In particular, we investigated the effect of high $(NH_4)_2SO_4$ concentrations on the proton-
promoted and oxalate-promoted Fe dissolution at low pH conditions. Our study also determined the Fe phases present in the
CFA and compared them to those present in mineral dust. The experimental results enabled us to develop a new Fe release
scheme for CFA sources which was then implemented into the global atmospheric chemical transport model IMPACT. The
model results were compared with observations of surface concentration of dissolved Fe in aerosol particles over the Bay of
Bengal from Bikkina et al. (2020).
**2 Materials and Methods**
**2.1 Sample collection and subsequent size fractionation**
CFA samples were collected from the electrostatic precipitators at three coal-fired power stations at different locations: United
Kingdom (Aberthaw ash), Poland (Krakow ash), and China (Shandong ash). The bulk samples were resuspended to obtain
dust fractions representative of particles emitted into the atmosphere. A custom-made resuspension system was used to collect
the $PM_{10}$ fraction (particles with an aerodynamic diameter smaller than 10 µm), which is shown in Fig. S1. Around 20 g of
sample was placed into a glass bottle and injected at regular intervals (2-5 sec) into a glass reactor (~70 L) by flushing the
bottle with pure nitrogen. The air in the reactor was pumped at a flow rate of 30 L min$^{-1}$ into a $PM_{10}$ sampling head.  Particles
were collected on 0.6 µm polycarbonate filters and transferred into centrifuge tubes. The system was cleaned manually and
flushed for 10 min with pure nitrogen before loading a new sample. A soil sample from Libya (Soil 5, 32.29237N/22.30437E)
was dry sieved to 63 µm (which is referred to as Libya dust) and used for the comparison of CFA with mineral dust.
**2.2 Fe dissolution kinetics**
The Fe dissolution kinetics of the CFA samples was determined by time-dependent leaching experiments. We followed a
similar methodology as in Ito and Shi (2016). $PM_{10}$ fractions were exposed to $H_2SO_4$ solutions at pH 1, 2 or 3, in the presence
of $H_2C_2O_4$ and/or $(NH_4)_2SO_4$ to simulate acidic processing in aerosol conditions. The concentration of $H_2C_2O_4$ in the
experiment solutions was chosen based on the molar ratio of oxalate and sulphate in $PM_{2.5}$ (particles with an aerodynamic
diameter smaller than 2.5 µm) from observations over the East Asia region (Yu et al., 2005). Around 50 mg of CFA was
leached in 50 ml of acidic solution to obtain a dust/liquid ratio of 1 g L$^{-1}$. The sample solution was mixed continuously on a
rotary mixer, in the dark at room temperature. A volume of 0.5 mL was sampled at fixed time intervals (2.5, 15, 60 min and 2,
6, 24, 48, 72, and 168 hours after the CFA sample was added to the experiment solution) and filtered through 0.2 µm pore size
syringe filters. The dissolved Fe concentration in the filtrate was determined using the ferrozine method (Viollier et al., 2000).
Leaching experiments were also conducted on the Libya dust. The relative standard deviation (RSD) at each sampling time
varied from 4 % to 15 % (n=7).
The pH of all the experiment solutions was calculated using the E-AIM model III for aqueous solutions (Wexler and Clegg,
2002). In part this was because the high ionic strength generated by the elevated concentration of $(NH_4)_2SO_4$ prevents
electrochemical sensors from making accurate pH measurements. For the experiment solutions with no $(NH_4)_2SO_4$, the pH
was measured by a pH meter before adding the ash and at the end of the experiments. The solution pH increased after adding
the ash, and the change in pH was used to estimate the buffer capacity of alkaline minerals in the samples, including for
example calcium carbonates ($CaCO_3$), lime (CaO), and portlandite ($Ca(OH)_2$). The estimated concentration of $H^+$ buffered was
used to input the concentration of $H^+$ into the E-AIM model. For each experiment, the pH was calculated before adding the
CFA samples and at the end of the experiments. The pH of the original solution before adding the samples was estimated from





the molar concentrations (mol L$^{-1}$) of $H_2SO_4$, $H_2C_2O_4$ and $(NH_4)_2SO_4$ used to prepare the solution. The model inputs included
the total concentrations of $H^+$ (without $H_2C_2O_4$ contribution), $NH_4^+$, $SO_4^{2-}$ and $H_2C_2O_4$. For the experiment solutions with no
$(NH_4)_2SO_4$, we calculated the final pH by reducing the total $H^+$ concentration input into the model to match the pH measured
at the end of the experiments. The buffered $H^+$ was then estimated from the difference between the original and final $H^+$
concentration input into the model. To determine the final pH of the solutions with high ionic strength, the $H^+$ concentration
input in the model was calculated as the difference between the $H^+$ concentration in the original solution and the buffered $H^+$
estimated at low ionic strength.
For the solution with no $(NH_4)_2SO_4$, the difference between calculated and measured pH is <7%. Table S1 reports the
concentrations of $H_2SO_4$, $H_2C_2O_4$ and $(NH_4)_2SO_4$ in the experiment solutions, the original and final pH from model estimates
(including $H^+$ concentrations and activities), and the pH measurements for the solution with low ionic strength.
**2.3 Sequential extractions**
The content of Fe oxide species in the samples was determined by Fe sequential extraction (Baldo et al., 2020; Poulton and
Canfield, 2005; Raiswell et al., 2008; Shi et al., 2011b). The Fe oxide species included highly reactive amorphous Fe oxide-
hydroxide (FeA), crystalline Fe oxide-hydroxide, mainly goethite and hematite (FeD), and Fe associated with magnetite (FeM).
To extract FeA, samples were leached in an ascorbate solution buffered at pH 7.5 (Raiswell et al., 2008; Shi et al., 2011b). The
ascorbate solution contained a deoxygenated solution of 50 g L$^{-1}$ sodium citrate, 50 g L$^{-1}$ sodium bicarbonate, and 10 g L$^{-1}$ of
ascorbic acid. Around 30 mg of CFA was leached for 24 hours in 10 mL of ascorbate extractant, mixed continuously on a
rotary mixer. The extraction solution was then filtered through a 0.2 µm membrane filter. In order to extract FeD, the residue
was leached for 2 more hours in a dithionite solution buffered at pH 4.8 (50 g L$^{-1}$ sodium dithionite in 0.35 M acetic acid and
0.2 M sodium citrate) (Raiswell et al., 2008; Shi et al., 2011b).
For the extraction of FeM, the CFA samples were first leached for 2 hours using a citrate-buffered dithionite solution to remove
FeD. The residue collected after filtration was then leached for 6 hours in a solution of 0.2 M ammonium oxalate $((NH_4)_2C_2O_4)$
and 0.17 M $H_2C_2O_4$ at pH 3.2 (Poulton and Canfield, 2005). The Fe extractions were all carried out in the dark at room
temperature. The Fe concentration in the filtered extraction solutions was measured using the ferrozine method (Viollier et al.,
2000) or by inductively coupled plasma optical emission spectrometry (ICP-OES) analysis for the solutions containing high
concentration of oxalate.
The total Fe content in the samples was determined by microwave digestion in concentrated nitric acid ($HNO_3$) followed by
inductively coupled plasma mass spectrometry (ICP-MS) analysis.
The RSD% obtained for each extract using the Arizona test dust was 3% for FeA, 11% for FeD, 12% for FeM and 2% for the
total Fe (n=7).
**2.4 X-ray absorption near edge structure (XANES) analysis**
We collected XANES spectra to qualitatively examine the Fe speciation in the CFA samples. The XANES spectra at the Fe
K-edge were collected at the Diamond Light Source beamline I18. A Si(111) double-crystal monochromator was used in the
experiments. The beam size was 400 µm×400 µm. THE XANES spectra were collected from 7000 to 7300 eV at a resolution
varying from 0.2 eV for 3 s in proximity to the Fe K-edge (7100–7125 eV) to 5 eV for 1 s from 7100 to 7300 eV. Powder
samples were suspended in methanol and deposited on Kapton® tape. The analysis was repeated three times. We measured the
XANES spectra of the CFA-PM$_{10}$ fractions and mineral standards including hematite, magnetite, and illite. Data were
processed using the Athena program, part of the software package Demeter (version 0.9.26) (Ravel and Newville, 2005).



## 2.5 Model description

This study used the Integrated Massively Parallel Atmospheric Chemical Transport (IMPACT) model (Ito et al., 2021a and
references therein). The model simulates the emission, chemistry, transport, and deposition of Fe-containing aerosols and the
precursor gases of inorganic and organic acids. The coating of acidic species on the surface of Fe-containing aerosols promotes
the release of soluble Fe in the aerosol deliquescent layer and enhances the aerosol Fe solubility (Li et al., 2017). On the other
hand, the external mixing of oxalate-rich aerosols with Fe-rich aerosols can suppress the oxalate-promoted Fe dissolution at
low concentration of oxalate near the source regions (Ito, 2015). However, the internal mixing of alkaline minerals such as
calcium carbonate with Fe-containing dust aerosols can suppress the Fe dissolution (Ito and Feng, 2010). Since CFA particles
are co-emitted with acidic species, the transformation of relatively insoluble Fe in coal combustion aerosols into dissolved Fe
is generally much faster than that for mineral dust aerosols during their atmospheric lifetime (Ito, 2015; Ito and Shi, 2016).
Additionally, the size of CFA particles is substantially smaller than that of mineral dust. Thus, we adopted an observationally
constrained parameter for the dry deposition scheme (Emerson et al., 2020) to improve the simulation of dry deposition velocity
of fine particles.
To improve the accuracy of our simulations of Fe-containing aerosols, we revised the on-line Fe dissolution schemes in the
original model (Ito et al., 2021a) in conjunction with the mineralogy-based emission rates and a more dynamic range of pH
estimates. To implement 3-step dissolution schemes, we used the mineral-specific emission inventory for anthropogenic Fe
emissions (Rathod et al., 2020). To apply the Fe dissolution schemes for high ionic strength in aerosols, we used the mean
activity coefficient for pH estimate (Pye et al., 2020). Moreover, the dissolution rate was assumed to be dependent of pH for
highly acidic solutions (pH < 2) unlike in the former dissolution scheme (Ito, 2015), which allowed us to predict the sensitivity
of Fe dissolution to pH lower than 2.
To validate the new dissolution scheme, we compared our model results with observations of surface concentration of dissolved
Fe in $PM_{2.5}$ aerosol particles over the Bay of Bengal (Bikkina et al., 2020).

## 3 Experimental results

### 3.1 Fe dissolution kinetics

We determined that Krakow ash had the largest buffer capacity, around 0.008 moles of buffered $H^+$ per litre, which was related
to the content of alkaline minerals in the sample. The buffer capacity of Aberthaw and Shandong ash was ~10 times smaller
than that of Krakow ash, around 0.0007 moles of buffered $H^+$ per litre. Leaching Krakow ash in 0.005 M $H_2SO_4$, the initial
concentration of $H^+$ was similar to the concentration of the $H^+$ buffered. As a result, the solution pH raised from
approximatively 2.1 to 2.7 corresponding to a pH change of around 20% (Table S1). For all the other experimental conditions,
the pH change was below 12% (Table S1). At the pH conditions used in this study (pH 1-3), acid buffering was fast and likely
occurred within the first 1-2 hours. We assumed that the calculated final pH was representative of the solution pH over the
duration of the experiments.
Dissolved Fe at different time intervals is reported as Fe%, which is the fraction of Fe dissolved to the total Fe content (FeT)
in the CFA samples. For all samples, a fast dissolution rate was observed at the beginning of the experiment. In the case of
Krakow ash, a dissolution plateau was reached after 2-hour leaching, which was likely due to the pH change. For that
sample/initial condition the pH increased to 2.7, and no more Fe was dissolved, leading to a total Fe solubility of ~9% over
the duration of the experiment (7 days) (Fig. 1a). Dissolving Krakow ash in 0.01 M $H_2SO_4$ (Fig. 1a), the experiment solution
had a final calculated pH of 2.1. The total Fe solubility was 34% at pH 2.1, almost 4 times higher than that at pH 2.7 (in 0.005
M $H_2SO_4$). Dissolution of Aberthaw and Shandong ash was slower compared to Krakow ash (Figs. 1b and 2c, respectively).



Leaching Aberthaw and Shandong ash in 0.005 M $H_2SO_4$ resulted in solutions with a pH of around 2.2. At this pH, the total
Fe solubility was 18% for Aberthaw ash and 21% for Shandong ash, which is 9-10 times higher than the total Fe solubility at
pH 2.9 (in 0.001 M $H_2SO_4$), around 2% for both samples.
The experimental treatment of dissolved Fe from Krakow ash in 0.05 $H_2SO_4$ solution with 1 M $(NH_4)_2SO_4$ (Fig. 1a) resulted
in a final predicted pH of 2.1. At that pH, the total Fe solubility of Krakow ash increased from 34% with no $(NH_4)_2SO_4$ to 48%
with high $(NH_4)_2SO_4$ concentration. The total Fe solubility of Krakow ash was around 28% at pH 3.0 with 1 M $(NH_4)_2SO_4$
(Fig. 1a), 3 times higher than that at pH 2.7 with no $(NH_4)_2SO_4$. At around pH 2, the total Fe solubility of Aberthaw (Fig. 1b)
and Shandong ash (Fig. 1c) increased by around 20% and 30% in the presence of $(NH_4)_2SO_4$. By contrast, the total Fe solubility
at pH 3.1 with 1 M $(NH_4)_2SO_4$ was 7.5% for Aberthaw ash (Fig. 1b) and 14% for Shandong ash (Fig. 1c), respectively, which
was around 4 and 7 times higher than in the experiments carried out at pH 2.9 without $(NH_4)_2SO_4$.
The Fe dissolution of the CFA samples in $H_2SO_4$ solutions with 0.01 M $H_2C_2O_4$ (at around pH 2) is shown in Fig. 2. The total
Fe solubility of Krakow ash at pH 1.9 with 0.01 M $H_2C_2O_4$ was 61% (Fig. 2a), which was almost 2 times higher than that at
pH 2.1 but without $H_2C_2O_4$ (Fig. 2a). For Aberthaw ash, oxalate contribution to the dissolution process led to a total Fe
solubility of 30% at pH 2.0 (Fig. 2b), which was 70% higher than in the experiment carried out in 0.005 M $H_2SO_4$ (~pH 2.2)
(Fig. 2b). Shandong ash dissolution behaviour was not affected by the presence of oxalate (Fig. 2c).
We also investigated the effect of high $(NH_4)_2SO_4$ concentration on oxalate-promoted dissolution. In Fig. 2a, the total Fe
solubility of Krakow ash decreased from 61% at pH 1.9 in the presence of oxalate to 54% at pH 2.0 with oxalate and $(NH_4)_2SO_4$.
For Aberthaw ash, the total Fe solubility at pH 2.0 decreased from 30% in the presence of oxalate to 19% after the addition of
$(NH_4)_2SO_4$ (Fig. 2b).
Figure 3 shows the Fe dissolution behaviour of Krakow ash at different pH conditions in the presence of 1 M $(NH_4)_2SO_4$ and
$H_2C_2O_4$ (0.01-0.03 M depending on the solution pH). The total concentration of oxalate ions was calculated using the E-AIM
model and was similar at different pH conditions, 0.015 at pH 1.0 (Experiment 7 Table S2), 0.009 at pH 2.0, and 0.01 at pH
2.9 (Experiments 3 Table S2). The highest total Fe solubility was observed at pH 1.0 (~67%). At pH 2.0, the total Fe solubility
decreased to 54%, and no substantial variations were observed between pH 2.0 and pH 2.9 (54%-51%). At pH 1.0, the
concentration of $H^+$ was considerably higher compared to pH 2.0-2.9, leading to a faster dissolution rate. The total
concentration of oxalate ions was 1.5-1.6 times higher in the solution at pH 1.0 than at pH 2.0-2.9, which may also contribute
to the faster dissolution rate. $C_2O_4^{-2}$ concentration increased with rising pH. Although the concentration of $H^+$ was lower at pH
2.9 than at pH 2.0, the E-AIM model estimated that $C_2O_4^{-2}$ contributed around 35% of the total oxalate concentration at pH
2.9, which was 4.5 times higher than at pH 2.0 (Experiments 3 Table S2). The similar dissolution behaviour at pH 2.0 and pH
2.9 conditions may reflect the combination of these two opposite factors, higher concentration of $C_2O_4^{-2}$ but lower
concentration of $H^+$ at pH 2.9 compared to 2.0.
We determined the Fe dissolution behaviour of Krakow ash at pH 1.0 in the presence of oxalate and increasing concentrations
of $(NH_4)_2SO_4$. The ash was leached in $H_2SO_4$ solutions with 0.03 M $H_2C_2O_4$ at pH 1.0, while the concentration of $(NH_4)_2SO_4$
varied from 0 to 1.5 M. In Fig. 4, the total Fe solubility of Krakow ash in the presence of oxalate was 75% at pH 1.0 and
decreased to 68% after the addition of 0.5 M $(NH_4)_2SO_4$. Higher $(NH_4)_2SO_4$ concentrations did not affect the Fe dissolution
behaviour in the presence of oxalate at pH 1.0.
**3.2 Fe speciation**
The Fe phases in the CFA samples determined through sequential extractions are shown in Fig. 5. The Fe speciation in the
Saharan dust sample is added for comparison. Krakow ash had a total Fe (FeT) content of 5.2%, while FeT in Aberthaw and

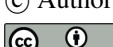



Shandong ash was 3.1% and 1.6% respectively. Amorphous Fe (FeA/FeT) was 6.5% in Krakow ash, 2% in Aberthaw ash, and
4.6% in Shandong ash. The CFA samples showed very different dithionite Fe (FeD/FeT) content, 21.5% in Krakow ash, 8%
in Aberthaw ash and 14.8% in Shandong ash. The content of magnetite (FeM/FeT) was considerably higher in Krakow ash
(22.4%) compared to Aberthaw (2.9%) and Shandong (4.5%) ash. About 50 %–87 % of Fe was contained in other phases most
likely in aluminosilicates. Overall, CFA had more magnetite and highly reactive amorphous Fe and less dithionite Fe than
Libya dust.
In Fig. S2, the Fe K-edge XANES spectra of Krakow and Aberthaw ash showed a single peak in the pre-edge region at around
7114.3 eV and 7114.6 eV, respectively. In the edge region, Aberthaw ash showed a broad peak at around 7132.2 eV, while the
peak of Krakow ash was slightly shifted to 7132.9 eV and narrower. The pre-edge peak at around 7115.4 suggest that Fe was
mainly as Fe(III). The spectral features of Aberthaw and Krakow ash are different from those of the hematite, magnetite and
illite standards suggesting that the glass fraction was dominant and controlled their spectral characteristics, which is consistent
with the results of the Fe sequential extractions. The XANES Fe K-edge spectra of the CFA samples have some common
features with those of Icelandic dust, but differs from northern African dust (Fig. S2). Aluminium silicate glass is also dominant
in Icelandic dust (Baldo et al., 2020). In the pre-edge region, Icelandic dust (sample MIR 45 in Fig. S2)  showed a main peak
at around 7114.4 eV and a second less intense peak at around 7112.7 eV, while a broad peak was observed at around 7131.9
eV in the edge region (Baldo et al., 2020). Northern African dust (western Sahara in Fig. S2) showed a distinct double peak in
the pre-edge region at around 7113.9 and 7115.2 eV, and a main peak in the edge region at around 7133.3 eV (Baldo et al.,

283   2020).

**4 Fe simulation from the IMPACT model**
**4.1 Fe dissolution scheme**
Based on the laboratory experiments carried out on the CFA samples, we implemented a 3-step dissolution scheme for proton-
promoted and oxalate-promoted Fe dissolution (Table 1). The Fe dissolution kinetics was described as follows (Ito, 2015):
$\sum_i RFe_i = k_i(pH, T) \times a(H^+)^{m_i} \times f_i$   (1)
where $RFe_i$ is the dissolution rate of individual mineral i, $k_i$ is the rate constant (moles Fe g$^{-1}$ s$^{-1}$), a(H$^+$) is the H$^+$ activity in
solution, $m_i$ represents the empirical reaction order for protons. The function $f_i$ (0 $\leq$ $f_i$ $\leq$1) accounts for the suppression of
mineral dissolution by competition for oxalate between surface Fe and dissolved Fe (Ito, 2015):
$f_i = 0.17 \times \ln([lig] \times [Fe]^{-1})_i + 0.63$   (2)
in which, [Fe] is the molar concentration (mol L$^{-1}$) of Fe$^{3+}$ dissolved in solution, and [lig] is the molar concentration of ligand
(e.g., oxalate). $f_i$ was set to 1 for the proton-promoted dissolution.
The scheme assumes 3 rate constants "fast", "intermediate" and "slow" for the proton-promoted, and the proton + oxalate-
promoted dissolution (Table 1). These were obtained by fitting the parameters to our measurements for Krakow ash in H$_2$SO$_4$
and (NH$_4$)$_2$SO$_4$ at pH 2-3, with and without oxalate (Experiments 2 and 3 in Table S1), which are shown in Fig. 6. The fast
rate constant represents highly reactive Fe species such as amorphous Fe oxyhydroxides, Fe carbonates and Fe sulphates. The
intermediate rate constant can be applied to nano-particulate Fe oxides, while more stable phases including for example Fe-
aluminosilicate and crystalline Fe oxides have generally slower rate (Ito and Shi, 2016; Shi et al., 2011a; Shi et al., 2011b; Shi
et al., 2015). Similarly, we predicted the dissolution kinetics of Aberthaw ash and Shandong ash (Figs. S3-S5). The dissolution





kinetic of Krakow ash was calculated based also on the experimental results at pH 1.0, which is shown in Fig. S6 in comparison
with kinetics predicted at pH 2.0 and pH 2.9 conditions.
The contribution of the oxalate-promoted dissolution to dissolved Fe was derived as the difference between the estimated
dissolution rates for the proton + oxalate-promoted dissolution and the proton-promoted dissolution:
$RFe_{i(oxalate)} = RFe_{i(proton + oxalate)} - RFe_{i(proton)}$ (3)
The Fe dissolution rates were predicted at a wider range of pH using Eq. (1) and Eq. (3) and the parameters in Table 1:
$RFe_i = RFe_{i(proton + oxalate)}$ when $RFe_{i(oxalate)} < 0$ (4)
Since $RFe_{i(oxalate)}$ is less than 0 at low pH (< 2), this equation applies to highly acidic conditions. As a result, the predicted
amount of dissolved Fe was smaller when using the dissolution rate for the proton + oxalate-promoted dissolution, $RFe_{i(proton + }$
$_{oxalate)}$, rather than the rate for the proton-promoted dissolution, $RFe_{i(proton)}$, at pH < 2. Accordingly, the dissolution rate, $RFe_i$,
was less dependent on the pH compared to $RFe_{i(proton)}$ at highly acidic conditions, possibly due to the competition for the
formation of surface complexes.
At pH > 2 when oxalate does promote Fe dissolution, the following equation applies:
$RFe_i = RFe_{i(proton)} + RFe_{i(oxalate)}$ when $RFe_{i(oxalate)} > 0$ (5)
**4.2 Surface concentration of dissolved Fe over the Bay of Bengal**
The new dissolution scheme was applied in the IMPACT atmospheric chemistry transport model to predict the surface
concentration of dissolved Fe in atmospheric particles collected over the Bay of Bengal, which is an area for which there are
detailed field measurements available (Bikkina et al., 2020; Kumar et al., 2010; Srinivas and Sarin, 2013; Srinivas et al., 2012)
and multi-modelling analyses have been done (Ito et al., 2019). It thus represents a test for our experimental results in actual
field conditions. Three sensitivity simulations were performed to explore the effects of the uncertainties associated with the
dissolution schemes and mineralogical component of Fe. In addition, the former setting (Ito et al., 2021a) was used in the
IMPACT model for comparison.
In sensitivity Test 0, the total Fe emission in anthropogenic aerosols was estimated using Fe emission factors by each sector
such as energy, heavy industry, and iron and steel industry for the simulation years (Ito et al., 2018), whereas in sensitivity
Test 1, Test 2, and Test 3, the mineral specific emission inventory for the year 2010 by Rathod et al. (2020) was used. In Test
0, we ran the model without the upgrades of the dissolution scheme discussed in section 2.4, and apply in addition the
photoinduced dissolution scheme for both combustion and dust aerosols (Ito, 2015; Ito and Shi, 2016), which was turned off
in Test 1, Test 2, and Test 3 due to the lack of laboratory measurements under high ionic strength. To estimate the aerosol pH,
we applied a $H^+$ activity coefficient of 1 for Test 0, while the mean activity coefficient from Pye et al. (2020) was used for the
other tests. The dissolution rate was assumed as pH-independent for highly acidic solutions (pH < 2) (Ito, 2015) in Test 0,
based on the laboratory measurements in Chen et al. (2012), while no pH threshold was considered in Test 1, Test 2, and Test
3 as the total dissolution (proton + oxalate) was suppressed at pH < 2 from the predicted dissolution rate.
In Test 1, we used the new dissolution scheme accounting for the proton- and oxalate- promoted dissolution of Krakow ash
for all combustion aerosols in the model (Table 1). The dissolution kinetics was calculated using the mineral-specific inventory
for anthropogenic Fe emissions (Rathod et al., 2020). The Fe composition of wood was used for open biomass burning (Matsuo
et al., 1992). In this simulation, 3 Fe pools were considered. Sulphate Fe in Rathod et al. (2020) was assumed as fast pool,



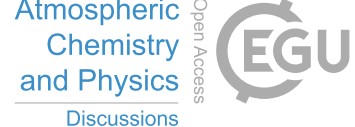

magnetite Fe as intermediate pool, hematite and Fe-aluminosilicate as slow pool. In Test 2, we calculated the dissolution
kinetics only considering the proton-promoted dissolution. In Test 3, the Fe pools were as determined here for Krakow ash:
ascorbate Fe (FeA) as fast pool, magnetite Fe (FeM) as intermediate pool, hematite plus goethite Fe (FeD) and other Fe as
slow pool (Fig. 5). FeA contains highly reactive Fe species with fast dissolution rates (Raiswell et al., 2008; Shi et al., 2011b).
FeM appeared to work well for the different fly ash samples in the dissolution scheme as intermediate Fe pool. FeD is
associated with crystalline Fe oxides and a predominant proportion of this is highly insoluble (Raiswell et al., 2008; Shi et al.,
2011b), thus it was considered as slow pool in the dissolution scheme. We assumed other Fe to be mostly as Fe-bearing
aluminosilicates and considered this as slow Fe pool.
The temporally and regionally averaged, model-calculated surface concentration of aerosol Fe (Fig. 7), dissolved Fe (Fig. 8)
and Fe solubility (Figs. 9 and S7) for the fine mode (PM$_{2.5}$) along the cruise tracks were compared with the measurements over
the Bay of Bengal for the period extending from 27 December 2008 to 26 January 2009 (Bikkina et al., 2020). The average
aerosol Fe concentration observed over the Bay of Bengal varies from $145 \pm 144$ ng m$^{-3}$ over the North Bay of Bengal (27
December 2008 - 10 January 2009) to $55 \pm 23$ ng m$^{-3}$ over the South Bay of Bengal (11-26 January 2009) (Bikkina et al.,
2020). In Fig. 7, the modelled aerosol Fe concentrations exhibit a similar variability to that of measurements with relatively
higher values over the North Bay of Bengal ($101 \pm 57$ ng m$^{-3}$ in Test 0, and $81 \pm 37$ ng m$^{-3}$ in Test 1-3) compared to the South
Bay of Bengal ($21 \pm 13$ ng m$^{-3}$ in Test 0, and $34 \pm 25$ ng m$^{-3}$ in Test 1- 3). The model reproduced the source apportion of Fe
(Fig. 7) which is qualitatively derived from previous observational studies indicating that the aerosol Fe concentrations over
the North Bay of Bengal are influenced by emissions of dust and combustion sources from the Indo-Gangetic Plain (Kumar et
al., 2010), whereas combustion sources (e.g., biomass burning and fossil-fuel) from South-East Asia are dominant over the
South Bay of Bengal (Kumar et al., 2010; Srinivas and Sarin, 2013). On the other hand, the model could not reproduce the
peak in total Fe concentration (1.8% of Fe content in PM$_{2.5}$ sample) reported around 29 December 2008. The total Fe observed
in PM$_{2.5}$ (613 ng m$^{-3}$) is higher than that in PM$_{10}$ (430 ng m$^{-3}$) (Srinivas et al., 2012). This may be due to the measurement
uncertainty including sample collection with two different high-volume samplers (Kumar et al., 2010).
The average aerosol dissolved Fe concentration measured over the North Bay of Bengal ($16 \pm 9$ ng m$^{-3}$) is slightly lower than
that over the South Bay of Bengal ($18 \pm 10$ ng m$^{-3}$) (Bikkina et al., 2020). The model prediction of dissolved Fe over the North
Bay of Bengal was $6 \pm 2$ ng m$^{-3}$ Fe in Test 0, $21 \pm 10$ ng m$^{-3}$ in Test 1, and $31 \pm 28$ ng m$^{-3}$ in Test 2, and $13 \pm 10$ ng m$^{-3}$ in
Test 3. The aerosol dissolved Fe estimated over the South Bay of Bengal was $6 \pm 1$ ng m$^{-3}$ in Test 0, $15 \pm 10$ ng m$^{-3}$ in Test 1,
$32 \pm 22$ ng m$^{-3}$ in Test 2, and $12 \pm 7$ ng m$^{-3}$ in Test 3. In Fig. 8, our model results show that the contribution of mineral dust
to aerosol dissolved Fe was higher over the North Bay of Bengal ($14\% \pm 6\%$ in Test 1, $28\% \pm 34\%$ in Test 2, and $33\% \pm 26\%$
in Test 3) compared to the South Bay of Bengal ($3\% \pm 1\%$ in Test 1, $1\% \pm 1\%$ in Test 2, and $3\% \pm 1\%$ in Test 3). Overall,
anthropogenic combustion sources were dominant over the Bay of Bengal accounting for $84\% \pm 12\%$ in Test 1, $72\% \pm 29\%$
in Test 2, and $69\% \pm 24\%$ in Test 3 of the aerosol dissolved Fe. Moreover, after 22 January 2009, the contribution of open
biomass burning sources increased up to 47% in Test 1, 64% in Test 2, and 60% in Test 3 (Fig. 8).
The aerosol Fe solubility measured over the South Bay of Bengal is higher than that over the North Bay of Bengal, respectively
$32\% \pm 11\%$ and $15\% \pm 7\%$ (Bikkina et al., 2020), and model estimates showed a similar trend (Fig. S7). In Fig. S7, the
calculated average Fe solubility over the North Bay of Bengal in Test 3 ($18\% \pm 10\%$) was in good agreement with observations,
while lower Fe solubility was estimated in Test 0 ($8\% \pm 5\%$) and higher values were obtained for Test 1 ($28\% \pm 8\%$). The
aerosol Fe solubility over the South Bay of Bengal was better captured in Test 1 ($43\% \pm 4\%$) and Test 3 ($39\% \pm 7\%$), whereas
Test 0 showed higher variability ($38\% \pm 22\%$). The proton-promoted dissolution scheme in Test 2 significantly overestimated
the Fe solubility over the Bay of Bengal (Figs. 9 and S7). The aerosol Fe solubility was largely overestimated in all scenarios
after 22 January 2009, as open biomass burning sources become dominant (Fig. 8). The comparison between observations and



model predictions of Fe solubility over the Bay of Bengal is shown in Fig. 9. The agreement between measurements and model
predictions was the best in Test 1 and Test 3. These exhibited good correlation with observations (R = 0.60 in Test 1 and R =
0.51 in Test 3), and the lowest centred root-mean-square (RMS) difference between the simulated and observed aerosol Fe
solubilities (RMS = 16 in Test 1 and RMS = 14 in Test 3). In Test 0, the model estimates showed higher difference from
observations (RMS = 22) and poor correlation (R = 0.30).

## 384   5 Discussion

### 385   5.1 Dissolution behaviour of Fe in CFA

In this study, the Fe dissolution kinetics of CFA samples from UK, Poland and China was investigated under simulated
atmospheric acidic conditions. A key parameter in both the atmosphere and the simulation experiments is the pH of the water
interacting with the CFA particles. The lower the pH of the experimental solution the faster the dissolution and eventually the
higher the amount of Fe dissolved. Our results showed a strong pH dependence in low ionic strength conditions, with higher
dissolution rate at lower pH. For example, reducing the solution pH from 2.7 to 2.1, the Fe solubility of Krakow ash increased
by a factor of 4 (Fig. 1a) over the duration of the experiments, while the Fe solubility of Aberthaw and Shandong ash increased
by 9-10 times from pH 2.9 to pH 2.2 (Figs. 1b-c). This enhancement is higher than that observed in studies conducted on
mineral dust samples, which showed that one pH unit can lead to 3-4 times difference in dissolution rates (Ito and Shi, 2016;
Shi et al., 2011a). Furthermore, Chen et al. (2012) reported that the Fe solubility of the certified CFA 2689 only increased by
10% from pH 2 to pH 1, after 50 hours of dissolution in acidic media. The Fe solubility of CFA ($PM_{10}$ fractions) after 6 hours
at pH 2 was 6%-10% for Aberthaw and Shandong ash respectively, and 28% for Krakow ash (Fig. 1). These values are higher
than the Fe solubilities measured by Fu et al. (2012), who reported 2.9%-4.2% Fe solubility in bulk CFA from three coal-fired
power plants in China after 12-hour leaching at pH 2. This suggest that Fe in our CFA samples initially dissolved faster than
those used in Fu et al. (2012). The Fe solubility after 72-hour leaching in $H_2SO_4$ at around pH 2 varied from around 12% and
17% (Aberthaw and Shandong ash) to 34% (Krakow ash). These values are at the lower end of the range or below those
reported in Chen et al. (2012), who measured a Fe solubility of ~20%-70% in certified CFA samples after accumulated acid
dissolution of 72 hours at pH 2. These results suggest that there are considerable variabilities in the pH dependent dissolution
of Fe in CFA.
Our results showed that high ionic strength has a major impact on dissolution rates of CFA at low pH (i.e., pH 2-3). The Fe
solubility of CFA increased by approximatively 20%-40% in the presence of 1 M $(NH_4)_2SO_4$ at around pH 2 over the duration
of the experiments, and by a factor from 3 to 7 at around pH 3 conditions (Fig. 1). At high ionic strength, the activity of ions
in solution is reduced, thus, in order to maintain similar pH conditions, the $H^+$ concentration has to be increased (Table S1).
Although Fe dissolution was primarily controlled by the concentration of $H^+$, the high concentration of sulphate ions could be
also an important factor contributing to Fe dissolution, in particular when the concentration of $H^+$ in the system was low (e.g.,
pH 3). Previous research found that the high ability of anions to form soluble complexes with metals can enhance Fe dissolution
(Cornell et al., 1976; Cornell and Schwertmann, 2003; Furrer and Stumm, 1986; Hamer et al., 2003; Rubasinghege et al., 2010;
Sidhu et al., 1981; Surana and Warren, 1969). Sulphate ions adsorbed on the particles surface form complexes with Fe (e.g.,
Rubasinghege et al., 2010). This may increase the surface negative charge favouring the absorption of $H^+$ and thereby increase
the dissolution rate. In addition, the formation of surface complexes may weaken the bonds between Fe and the neighbouring
ions (Cornell et al., 1976; Furrer and Stumm, 1986; Sidhu et al., 1981). Cwiertny et al. (2008) reported that at pH 1-2 the high
ionic strength generated by NaCl up to 1 M did not influence Fe dissolution of mineral dust particles. However, Ito and Shi
(2016) showed that the high ionic strength resulting from the addition of 1 M $(NH_4)_2SO_4$ in leaching solutions at pH 2-3
enhanced the Fe dissolution of dust particles, which was also observed here for the CFA samples. Borgatta et al. (2016)





compared the Fe solubility of CFA from USA Midwest, North-East India, and Europe in acidic solution (pH 1-2) containing
1 M NaCl. The Fe solubility measured after 24 hours varied from 15% to 70% in different CFA (bulk samples) at pH 2 with 1
M NaCl, which was considerably higher than that observed at pH 2 with 1 M NaNO$_3$ (<20%) (Kim et al., 2020). Both studies
did not investigate the impact of ionic strength on the dissolution behaviour, i.e., by comparing the dissolution at low and high
ionic strength. Note that both studies did not specify how the pH conditions were maintained at pH 2. Here, we considered the
most important sources of high ionic strength in aerosol water and simulated Fe dissolution in the presence of $(NH_4)_2SO_4$ and
$H_2C_2O_4$ under acidic conditions. We emphasize that the pH under high ionic strength here is estimated from a thermodynamic
model, similar to those implemented in the IMPACT model.
The presence of oxalate enhanced Fe dissolution in Krakow and Aberthaw ash but not in Shandong ash at around pH 2 (Fig.
2). The effect of oxalate on the Fe dissolution kinetics has also been studied by Chen and Grassian (2013) at pH 2 (11.6 mM
$H_2C_2O_4$). After 45-hour leaching, the Fe solubility of the certified CFA 2689 increased from 16% in $H_2SO_4$ at pH 2 to 44% in
$H_2C_2O_4$ at the same pH (Chen and Grassian, 2013). Therefore, the enhancement in Fe solubility of CFA in the presence of
oxalate observed in this study (from no impact in Shandong ash to doubled dissolution in Krakow ash) is lower than that
reported for the certified CFA 2689 which was around by 2.8 times (Chen and Grassian, 2013). Since no data are available in
Chen and Grassian (2013), we are unable to make a comparison with the other two certified CFA samples. The Fe solubility
of Krakow ash after 48-hour leaching at pH 1.9 with 0.01 M $H_2C_2O_4$ (Fig. 2a) was 53%, which is within the range observed
in Chen and Grassian (2013) for the certified CFA samples at similar pH and $H_2C_2O_4$ concentrations (from 44% to 78%),
whereas the Fe solubility of Aberthaw and Shandong ash (Figs. 2b-c, 18%-17% after 48-hour leaching at pH 2.0 with 0.01 M
$H_2C_2O_4$) was considerably lower than that of certified CFA (Chen and Grassian, 2013). These results suggest a large variability
in the effects of oxalate on the Fe dissolution rates in different types of CFA.
Our results also indicated that high $(NH_4)_2SO_4$ concentrations suppress oxalate-promoted Fe dissolution of CFA (Fig. 2), which
was not considered in previous research. At pH 1.9 in the presence of oxalate, the Fe solubility of Krakow ash decreased by
around 10% after the addition of $(NH_4)_2SO_4$, while the Fe solubility of Aberthaw ash decreased by 35% (Fig. 2). We used the
E-AIM model to estimate the concentration of oxalate ions and their activity (Table S2). The pH influences the speciation of
$H_2C_2O_4$ in solution (e.g., Lee et al., 2007). $H_2C_2O_4$ is the main species below pH 2, whereas $HC_2O_4^-$ is dominant between pH
1-4. Above pH 4, $C_2O_4^{-2}$ is the principal species. In our experiments, $H_2C_2O_4$ is mainly as $HC_2O_4^-$ at around pH 2 (Experiments
3-4 in Table S2). In the presence of $(NH_4)_2SO_4$, the activity coefficient of $HC_2O_4^-$ was reduced by approximatively 35-38%
(Experiments 3 in Table S2). Increasing the ionic strength lowers the activity of the oxalate ions, but at the same time favours
the dissociation of the acid. At around pH 2 conditions, the E-AIM model estimated that the activity of $C_2O_4^{-2}$ was reduced by
around one order of magnitude in the presence of $(NH_4)_2SO_4$, while its concentration increased 12-15 times (Experiments 3 in
Table S2). The adsorption of anions can reduce oxalate adsorption on the particle surface due to electrostatic repulsion which
results in slower dissolution rates (Eick et al., 1999). Precipitation of ammonium hydrogen oxalate ($NH_4HC_2O_4$) can also occur
in the system, but this is very soluble and easily re-dissolves forming soluble oxalate species (Lee et al., 2007). We speculate
that the high concentration of sulphate ions is likely to be responsible for inhibiting the oxalate-promoted dissolution by
reducing oxalate adsorption on the particle surface. At pH 1 in the presence of oxalate, increasing the concentration of
$(NH_4)_2SO_4$ from 0.5 M to 1.5 M did not affect the Fe dissolution behaviour of the CFA samples (Fig. 4). As previously
discussed, the adsorption of sulphate ions on the particle surface may inhibit oxalate-promoted dissolution. However, once the
saturation coverage is reached, increasing the concentration of anions has no further effect on the dissolution rate (Cornell et
al., 1976).
Fe speciation is an important factor affecting the Fe dissolution behaviour. CFA particles have very different chemical and
physical properties depending for example on the nature of coal burned, combustion conditions, cooling process and particle





control devices implemented at the power stations (e.g., Blissett and Rowson, 2012; Yao et al., 2015). This is likely the reason
why the Fe speciation observed in the CFA samples analysed in this study from different location varied considerably (Fig. 5).
In the CFA samples, the Fe dissolution curves for different pH and ionic strengths generally showed the greatest rate of Fe
release within the first 2 hours, followed by a slower dissolution, reaching almost a plateau at the end of the experimental run.
This indicates the presence of multiple Fe phases in CFA particles with a wide range of reactivity. Initially, highly reactive
phases were the main contribution to dissolved Fe. As the dissolution continued, more stable phases became the dominant
source of dissolved Fe (Shi et al., 2011a). SEM analysis conducted on CFA samples showed that CFA particles are mostly
spherical (e.g., Chen et al., 2012; Dudas and Warren, 1987; Valeev et al., 2018; Warren and Dudas, 1989) with Fe oxide
aggregates on the surface (Chen et al., 2012; Valeev et al., 2018). The analysis of the CFA samples processed in aqueous
solution at low pH suggests that initially Fe dissolved from the reactive external glass coating (Dudas and Warren, 1987;
Warren and Dudas, 1989) and from the Fe oxide aggregates on the particle surface (Chen et al., 2012; Valeev et al., 2018).
Subsequently, Fe is likely realised from the structure of the aluminium silicate glass (Chen et al., 2012; Dudas and Warren,
1987; Valeev et al., 2018; Warren and Dudas, 1989), and crystalline Fe oxide phases (Warren and Dudas, 1989). Overall,
Krakow ash showed the fastest dissolution rates, but the dissolution of highly reactive Fe species as FeA is insufficient to
account for the high Fe solubility observed at low pH. Our results showed that once the FeA dissolved, additional Fe was
dissolved from more refractory Fe phases. The modelled dissolution kinetics obtained using FeM as intermediate pool were in
good agreements with measurements (Figs. S3-S6). FeM is likely to be primary magnetite but may contain a fraction of the
more reactive aluminosilicate glass. Our model results suggest that magnetite in CFA particles may be more soluble than has
been shown in Marcotte et al. (2020). It is possible that in real CFA samples the mineral physicochemical properties including
for example crystal size, degree of crystallinity, cationic and anionic substitution in the lattice which influence the Fe
dissolution behaviour (e.g., Schwertmann, 1991) are likely to be different from those of the reference minerals analysed in
Marcotte et al. (2020). In order to estimate in detail the relative contribution of different mineral phases to dissolved Fe, most
detailed work would be needed to determine Fe mineral phases in pristine and processed CFA particles.
Finally, the modelled dissolution kinetics obtained using the new dissolution scheme for CFA (Table 1) showed better
agreement with laboratory measurements than when using the original scheme (Ito, 2015) (Figs. S8 and S9). In Fig. S8, we
compared the Fe dissolution kinetics of Krakow ash at around pH 2 and 3 with 1 M $(NH_4)_2SO_4$ calculated using the proton-
promoted dissolution scheme in Table 1 with the dissolution kinetics calculated at similar pH but using the proton-promoted
dissolution scheme for combustion aerosols in Ito (2015) (Table S3). The dissolution scheme in Ito (2015) was based on
laboratory measurements conducted at low ionic strength (Chen et al., 2012) and assumed a single Fe phase in combustion
aerosol particles, while the new dissolution scheme considered the high ionic strength of aerosol water and assumed three rate
constants, for fast, intermediate and slow kinetics of the different Fe phases present in CFA particles. The Fe dissolution
kinetics obtained using the new dissolution scheme showed a better agreement with measurements and was enhanced compared
to the model estimates obtained using the original dissolution scheme (Ito, 2015) for low ionic strength conditions (Fig. S8).
Figure S9 shows the Fe dissolution kinetics of Krakow ash at pH 2.0 and 2.9 with 0.01 M $H_2C_2O_4$ and 1 M $(NH_4)_2SO_4$
calculated using the proton- and oxalate-promoted dissolution scheme in Table 1 and the dissolution kinetics calculated at
similar pH and $H_2C_2O_4$ concentration but using the scheme in Ito (2015) (i.e., single phase dissolution, see Table S3). The Fe
dissolution kinetics predicted using the new dissolution scheme had a much better agreement with measurements. Figure S9c
shows the suppression of the oxalate-promoted dissolution at pH 2.0 and high $(NH_4)_2SO_4$ concentrations. At pH 2, the proton-
promoted dissolution was comparable to the proton + oxalate-promoted dissolution (Fig. S9c), with $RFe_{(oxalate)}$ close to zero
(see Eq. 3). At pH 2.9, the proton + oxalate-promoted dissolution was higher than the proton + oxalate-promoted dissolution
(Fig. S9d), with $RFe_{(oxalate)} > 0$ (Eq. 5).





Moreover, the new 3-step dissolution scheme better captured the initial fast dissolution of CFA (Figs. 2-3) which was also
observed in previous research (Borgatta et al., 2016; Chen et al., 2012; Chen and Grassian, 2013; Fu et al., 2012; Kim et al.,
2020) (except for the certified CFA 2689 in Chen et al. (2012) which showed increasing dissolution rates over the duration of
the experiment). Furthermore, the new scheme enabled us to account for the different Fe speciation determined in the CFA
samples, which could be a key factor contributing to the different Fe dissolution behaviour observed in the present study and
in literature (Borgatta et al., 2016; Chen et al., 2012; Chen and Grassian, 2013; Fu et al., 2012; Kim et al., 2020). In Figs. S3-
S5, the dissolution kinetics of Aberthaw and Shandong ash calculated using the dissolution rates in Table 1 and the Fe phases
determined in the samples showed a good agreement with measurements.
**5.2 Comparison with mineral dust**
High ionic strength also impacted the dissolution rates of the Saharan dust sample at low pH (Fig. S10). At around pH 2
conditions, the proton-promoted Fe dissolution of Libya dust was enhanced by ~40% after the addition of $(NH_4)_2SO_4$. At
around pH 2 and with 0.01 M $H_2C_2O_4$, the Fe solubility of Libya dust decreased by ~30% in the presence of $(NH_4)_2SO_4$.
Overall, the Fe solubility of Libya dust was lower compared to that observed in the CFA samples. After 168 hour-leaching at
pH 2.1 with 1 M $(NH_4)_2SO_4$, the Fe solubility of Libya dust was 7.2% (Fig. S10), which was from around 3 to 7 times lower
compared to that of the CFA samples (Fig. 1). At around pH 2 conditions in the presence of oxalate and high $(NH_4)_2SO_4$
concentration, the Fe solubility of Libya dust rose to ~13.6% (Fig. S10), which is still 4 times lower than that of Krakow ash
and around 1.5 lower than Aberthaw and Shandong ash (Fig. 2). The Fe solubilities of Libya dust observed in this study are
comparable with those of the Tibesti dust in Ito and Shi (2016) at similar experimental conditions.
The enhanced Fe solubility in CFA compared to mineral dust could be primarily related to the different Fe speciation (Figs. 5
and S2). CFA contained more highly reactive Fe and magnetite but less hematite and goethite than mineral dust.
Although mineral dust is the largest contribution to aerosol Fe while CFA accounts for only a few percent, atmospheric
processing of CFA may result in a larger than expected contribution of bioavailable Fe deposited to the surface ocean. It is
thus important to quantify the amount and nature of CFA in atmospheric particles.
**5.3 Comparison of modelled Fe solubility with field measurements**
The model results obtained using the emission inventory from Rathod et al. (2020) and the new dissolution scheme for the
proton + oxalate-promoted dissolution (Table 1) in Test 1 and Test 3 provided a better estimate of dissolved Fe over the Bay
of Bengal than the other tests (Figs. 8, 9, and S7). At the same time, the new model improved the agreement of aerosol Fe
solubility from Test 0 (70% ± 4%) to Test 1 (44% ± 3%) and Test 3 (48% ± 1%) with the field data (25% ± 3%) but still
overestimated it after 22 January 2009, when open biomass burning sources become dominant (Fig. 8). This could be due to
the unrepresentative Fe speciation used in Test 1 and Test 3 for biomass burning over the Bay of Bengal. To reduce the
uncertainty in model predictions, emission inventories could be improved through a comprehensive characterization of Fe
species in combustion aerosol particles.
The revised model also enabled us to predict sensitivity to a more dynamic range of pH changes, particularly between
anthropogenic combustion and biomass burning. The results show that the proton-promoted dissolution scheme in Test 2
significantly overestimated aerosol dissolved Fe (Figs. 8, 9 and S7), which indicates the suppression of the proton + oxalate-
promoted dissolution at pH < 2. In Fig. 10, the model estimates of surface concentration of dissolved Fe over the Bay of Bengal
considerably improved in Test 1 compared to Test 0. The model results in Test 1 also indicate a larger contribution of pyrogenic
dissolved Fe over regions with strong anthropogenic source such as East Asia, but a smaller contribution downwind from
tropical biomass burning regions (Fig. 11). We demonstrated that the implementation of the new Fe dissolution scheme,





including a rapid Fe release at the initial stage and highly acidic conditions, enhanced the model estimates. However, in Test 1, we turned off the photo-reductive dissolution scheme (Ito, 2015), which was based on the laboratory measurements in Chen and Grassian (2013). To determine the photoinduced dissolution kinetic of CFA particles it is necessary to account for the effect of high concentration of $(NH_4)_2SO_4$ on photo-reductive dissolution rate which should be considered in future research.

**Data availability statement**

The new dissolution schemes for the proton-promoted and oxalate-promoted dissolution are reported in Table 1. Table S1 reports the concentrations of $H_2SO_4$, $H_2C_2O_4$ and $(NH_4)_2SO_4$ in the experiment solutions, the original and final pH from model estimates (including $H^+$ concentrations and activities), and the pH measurements for the solution with low ionic strength. Table S2 contains the summary of the concentration and activity of total oxalate ions, $C_2O_4^{2-}$, and $HC_2O_4^-$ in the experiment solutions calculated using the E-AIM model III. The observations of the surface concentration of aerosol Fe, dissolved Fe and Fe solubility for the fine mode ($PM_{2.5}$) over the Bay of Bengal are from Bikkina et al. (2020) and are available at https://pubs.acs.org/doi/10.1021/acsearthspacechem.0c00063. The Fe speciation, the measurements of the Fe dissolution kinetic, and the results of the IMPACT model for each sensitivity simulation (Test 0-3) can be downloaded at: https://doi.org/10.25500/edata.bham.00000702.

**Author contributions**

CB, ZS, and AI designed the experiments and discussed the results. ZS supervised the experimental and data analyses. CB conducted the experiments and the data analysis with contributions from ZS, AI, MDK and ND. ND, ZS and KI performed the XANES measurements. AI developed the model of the dissolution kinetics and performed the model simulations. Krakow and Aberthaw ash were provided by TJ, while Shandong ash was provided by WL. Soil 5 from Libya was collected by ND. CB prepared the article with contributions from MDK and all the other co-authors.

**Competing interests**

The authors declare that they have no conflict of interest.

**Acknowledgments**

CB is funded by the Natural Environment Research Council (NERC) CENTA studentship (grant no. NE/L002493/1). Support for this research was provided to AI by JSPS KAKENHI (grant no. 20H04329), Integrated Research Program for Advancing Climate Models (TOUGOU) (grant no. JPMXD0717935715) from the Ministry of Education, Culture, Sports, Science and Technology (MEXT), Japan. We acknowledge Diamond Light Source for time on Beamline/Lab I18 under the Proposals: SP22244-1; SP12760-1; SP10327-1.

**Financial support**

This research has been supported by the Natural Environment Research Council (grant no. NE/L002493/1), JSPS KAKENHI (grant no. 20H04329), the Integrated Research Program for Advancing Climate Models (TOUGOU) (grant no. JPMXD0717935715).





**Table 1. Constants used to calculate Fe dissolution rates for fossil fuel combustion aerosols, based on laboratory experiments**
**conducted at high ionic strength.**

| Stage | Kinetic | Scheme | Rate constant - k(pH, T)[a] | m[c] |
|---|---|---|---|---|
| I | Fast | Proton | $7.61 \times 10^{-6}\exp[E(pH)^{b} \times (1/298 - 1/T)]$ | 0.241 |
| II | Intermediate | Proton | $1.91 \times 10^{-7}\exp[E(pH)^{b} \times (1/298 - 1/T)]$ | 0.195 |
| III | Slow | Proton | $2.48 \times 10^{-7}\exp[E(pH)^{b} \times (1/298 - 1/T)]$ | 0.843 |
| I | Fast | Proton + Oxalate | $5.54 \times 10^{-6}\exp[E(pH)^{b} \times (1/298 - 1/T)]$ | 0.209 |
| II | Intermediate | Proton + Oxalate | $1.50 \times 10^{-7}\exp[E(pH)^{b} \times (1/298 - 1/T)]$ | 0.091 |
| III | Slow | Proton + Oxalate | $1.77 \times 10^{-8}\exp[E(pH)^{b} \times (1/298 - 1/T)]$ | 0.204 |

[a] k(pH, T) is the pH and temperature dependent 'far-from-equilibrium' rate constant (moles Fe $g^{-1}$ $s^{-1}$). The Fe dissolution
scheme assumes 3 rate constants "fast", "intermediate" and "slow" for the proton- and oxalate-promoted dissolution. The
parameters were fit to our measurements for Krakow ash.
[b] $E(pH) = -1.56 \times 10^{3} \times pH + 1.08 \times 10^{4}$. The parameters were fit to the measurements for soils (Bibi et al., 2014).
[c] m is the reaction order with respect to aqueous phase protons, which was determined by linear regression from our
experimental data in the pH range between 2 and 3 for proton- and oxalate-promoted dissolution schemes.

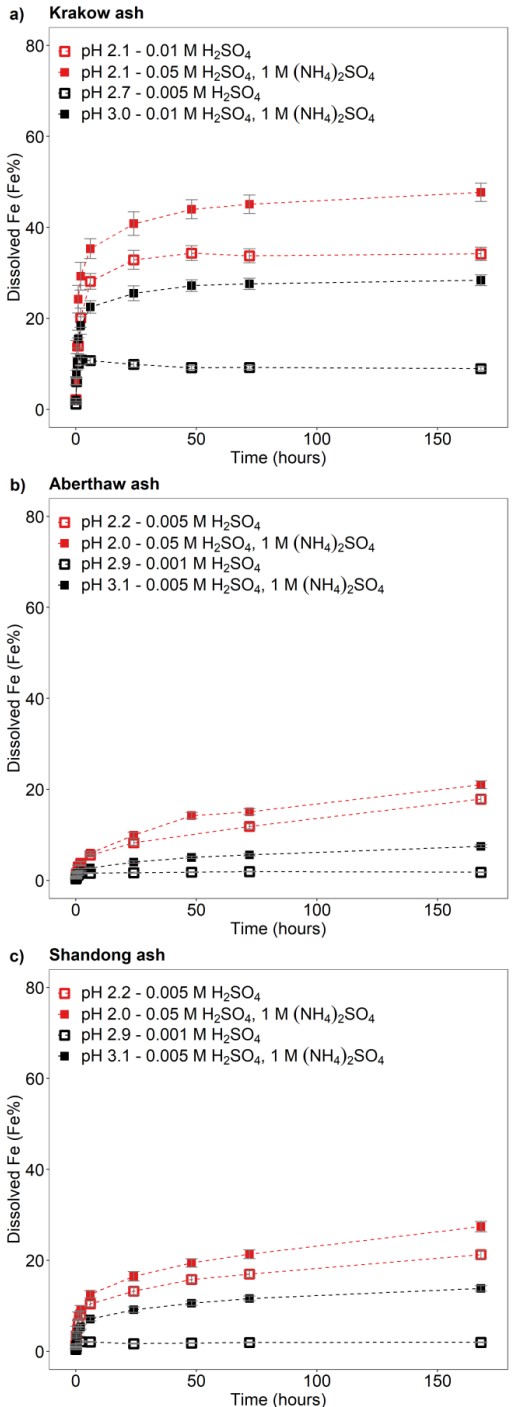


**Figure 1: Fe dissolution kinetics of a) Krakow ash, b) Aberthaw ash and c) Shandong ash in H2SO4 solutions (open rectangles) and with 1 M (NH$_4$)$_2$SO$_4$ (filled rectangles). The molar concentrations of H$_2$SO$_4$ and (NH$_4$)$_2$SO$_4$ in the experiment solutions are shown. The final pH of the experiment solutions is also reported, which was calculated using the E-AIM model III for aqueous solution (Wexler and Clegg, 2002) accounting for the buffer capacity of the CFA samples (Experiments 1-2 in Table S1). The experiments conducted at around pH 2 are in red, while the experiments at around pH 3 are in black. The data uncertainty was estimated using the error propagation formula.**



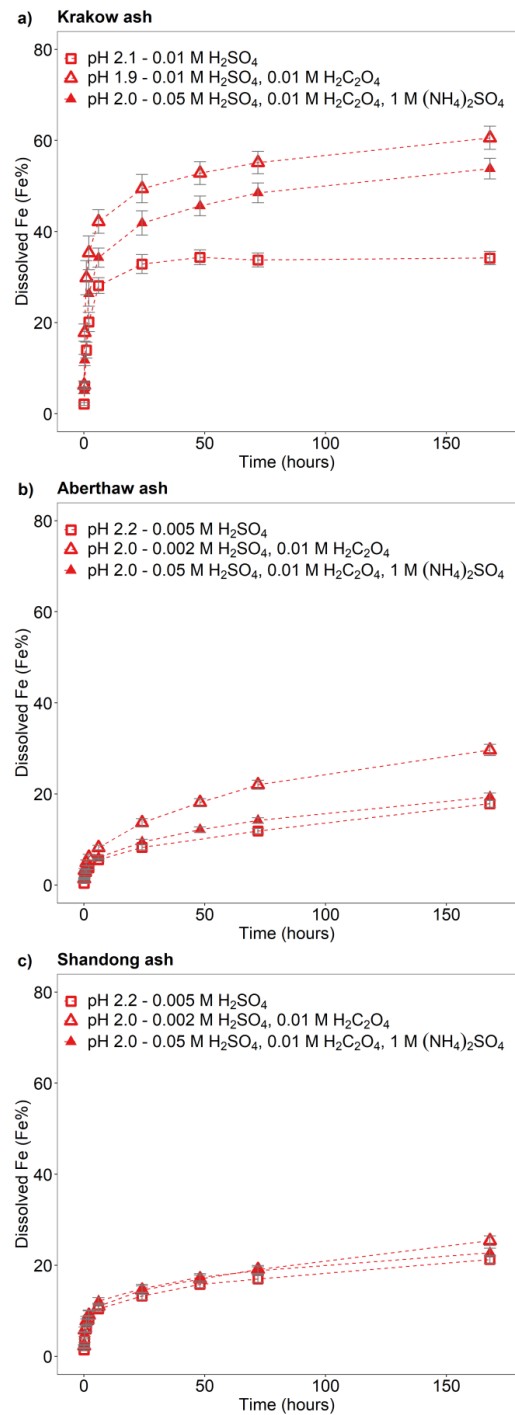

Figure 2: Fe dissolution kinetics of a) Krakow ash, b) Aberthaw ash, and c) Shandong ash in $H_2SO_4$ solutions at around pH 2 (red open rectangles), with 0.01 M $H_2C_2O_4$ (red open triangles), and 1 M $(NH_4)_2SO_4$ (red filled triangles). The molar concentrations of $H_2SO_4$, $H_2C_2O_4$ and $(NH_4)_2SO_4$ in the experiment solutions are shown. The final pH of the experiment solutions is also reported, which was calculated using the E-AIM model III for aqueous solution (Wexler and Clegg, 2002) accounting for the buffer capacity of the CFA samples (Experiments 1, 3-4 at around pH 2). The data uncertainty was estimated using the error propagation formula.





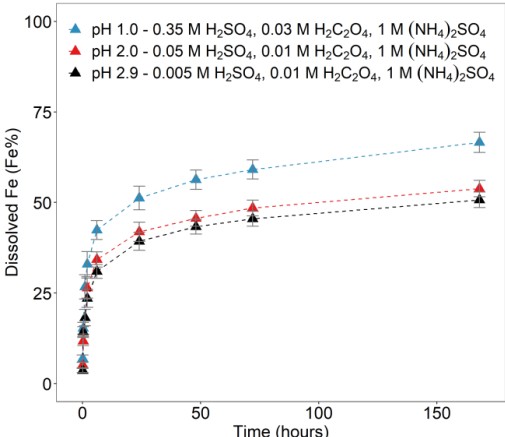

594

**Figure 3: Fe dissolution kinetics of Krakow ash in H₂SO₄ solutions at pH 1.0 with 0.03 M H₂C₂O₄ and 1 M (NH₄)₂SO₄ (blue filled triangles), at pH 2.0 with 0.01 M H₂C₂O₄ and 1 M (NH₄)₂SO₄ (red filled triangles), and at pH 2.9 with 0.01 M H₂C₂O₄ and 1 M (NH₄)₂SO₄ (black filled triangles). The molar concentrations of H₂SO₄, H₂C₂O₄ and (NH₄)₂SO₄ in the experiment solutions are shown. The final pH of the experiment solutions is also reported, which was calculated using the E-AIM model III for aqueous solution (Wexler and Clegg, 2002) accounting for the buffer capacity of the CFA samples (Experiment 7 at pH 1.0, Experiment 3 at pH 2.0, and Experiment 3 at pH 2.9 in Table S1). The data uncertainty was estimated using the error propagation formula.**

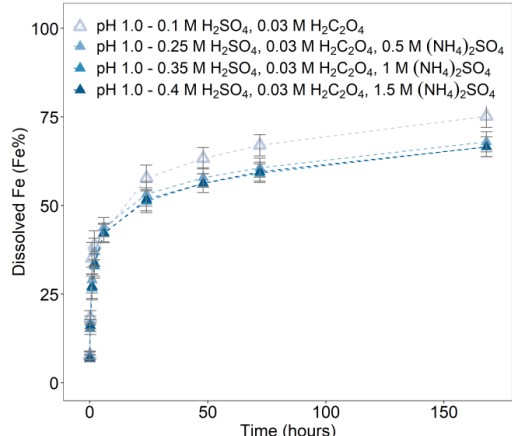

601

**Figure 4: Fe dissolution kinetics of Krakow ash in H₂SO₄ solutions at pH 1.0 with 0.03 M H₂C₂O₄ and concentration of (NH₄)₂SO₄ from 0 to 1.5 M. The molar concentrations of H₂SO₄, H₂C₂O₄ and (NH₄)₂SO₄ in the experiment solutions are shown. The final pH of the experiment solutions is also reported, which was calculated using the E-AIM model III for aqueous solution (Wexler and Clegg, 2002) accounting for the buffer capacity of the CFA samples (Experiments 5-8 in Table S1). The data uncertainty was estimated using the error propagation formula.**



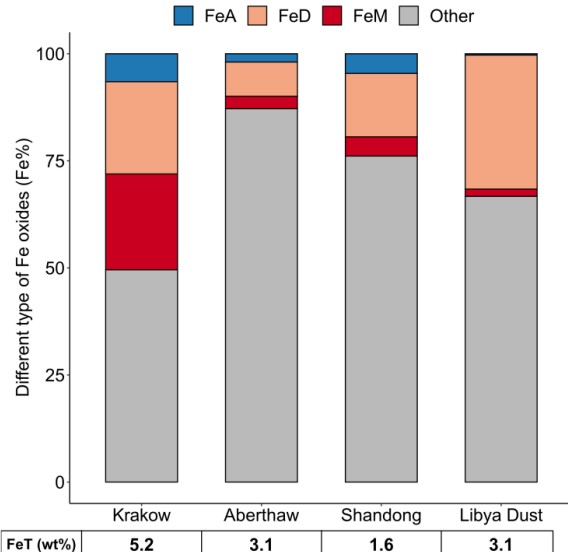

607

**Figure 5: Percentages of ascorbate Fe (FeA), dithionite Fe (FeD), magnetite Fe (FeM) and other Fe to the total Fe (FeT) in the coal fly ash samples and mineral dust from Africa (Libya dust). The FeT (as %wt.) was given below each sample column. The data uncertainty was estimated using the error propagation formula: 4% for FeA/FeT, 11% for FeD/FeT, 12% for FeM/FeT, and 2 % for FeT.**

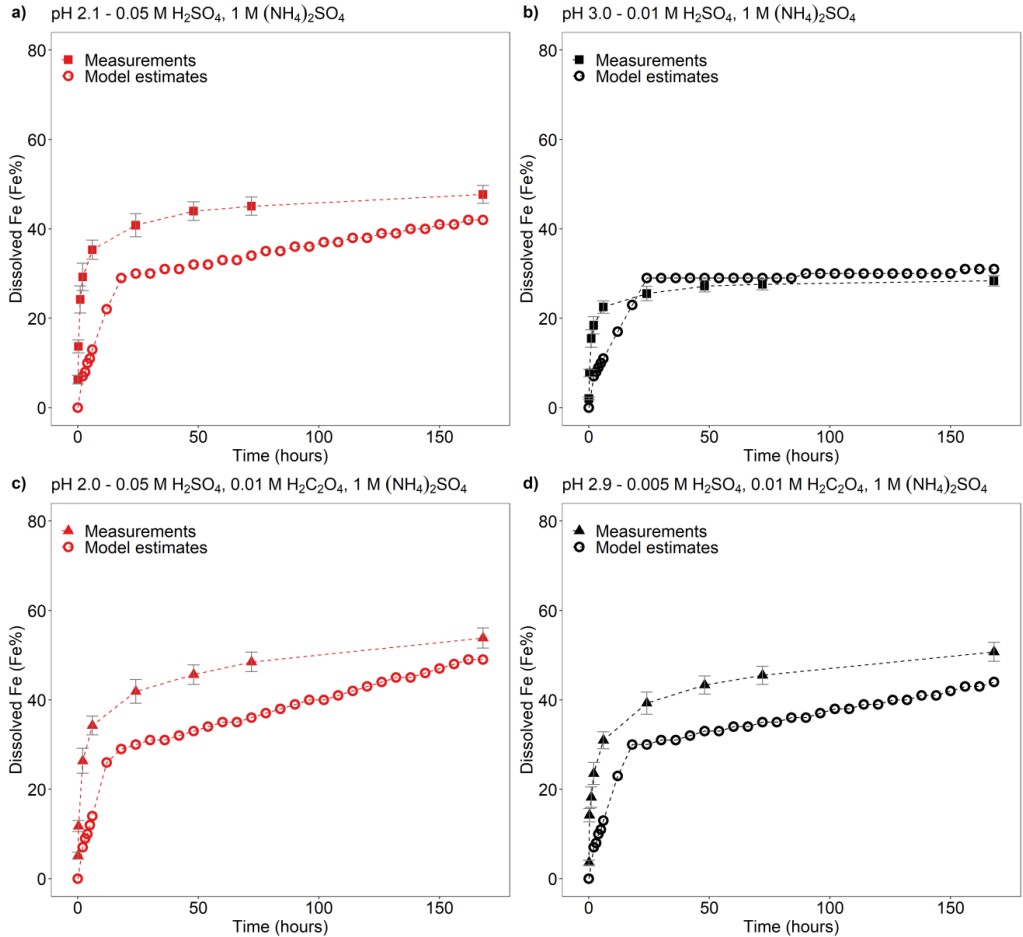

**Figure 6: Comparison between the Fe dissolution kinetics of Krakow ash predicted using Eq. (1) and measured in $H_2SO_4$ solutions a-b) with 1 M $(NH_4)_2SO_4$, c-d) with 0.01 M $H_2C_2O_4$ and 1 M $(NH_4)_2SO_4$. The molar concentrations of $H_2SO_4$, $H_2C_2O_4$ and $(NH_4)_2SO_4$ in the experiment solutions are shown. The final pH of the experiment solutions is also reported, which was calculated using the E-AIM model III for aqueous solution (Wexler and Clegg, 2002) accounting for the buffer capacity of the CFA samples (Experiments 2-3 in Table S1). The experiments conducted at around pH 2 are in red, while the experiments at around pH 3 are in black. The data uncertainty was estimated using the error propagation formula.**



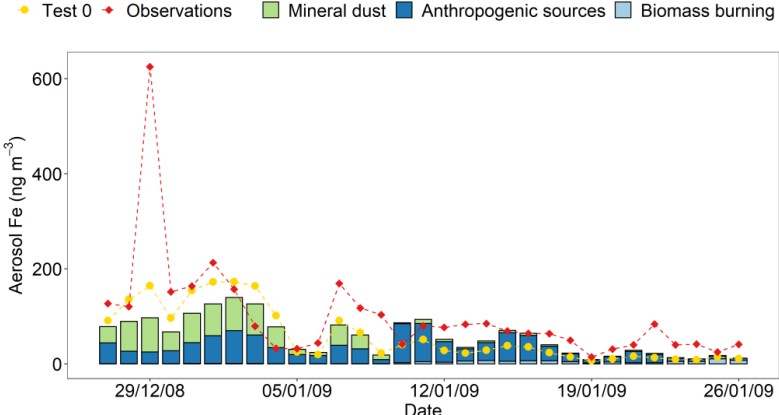

619

**Figure 7: Surface concentration of Fe in PM$_{2.5}$ aerosol particles over the Bay of Bengal from 27 December 2008 to 26 January 2009. Observations are from Bikkina et al. (2020) (red filled diamonds). Aerosol Fe was calculated along the cruise tracks using the IMPACT model. The total Fe emission in anthropogenic aerosols was estimated using Fe emission factors by each sector such as energy, industry, and iron and steel industry for the simulation years (Ito et al., 2018) in sensitivity Test 0 (yellow filled circles), while the mineral specific emission inventory for the year 2010 by Rathod et al. (2020) was used in the other tests. The contribution of mineral dust sources, anthropogenic sources and biomass burning to total Fe is shown for Test 1-3.**





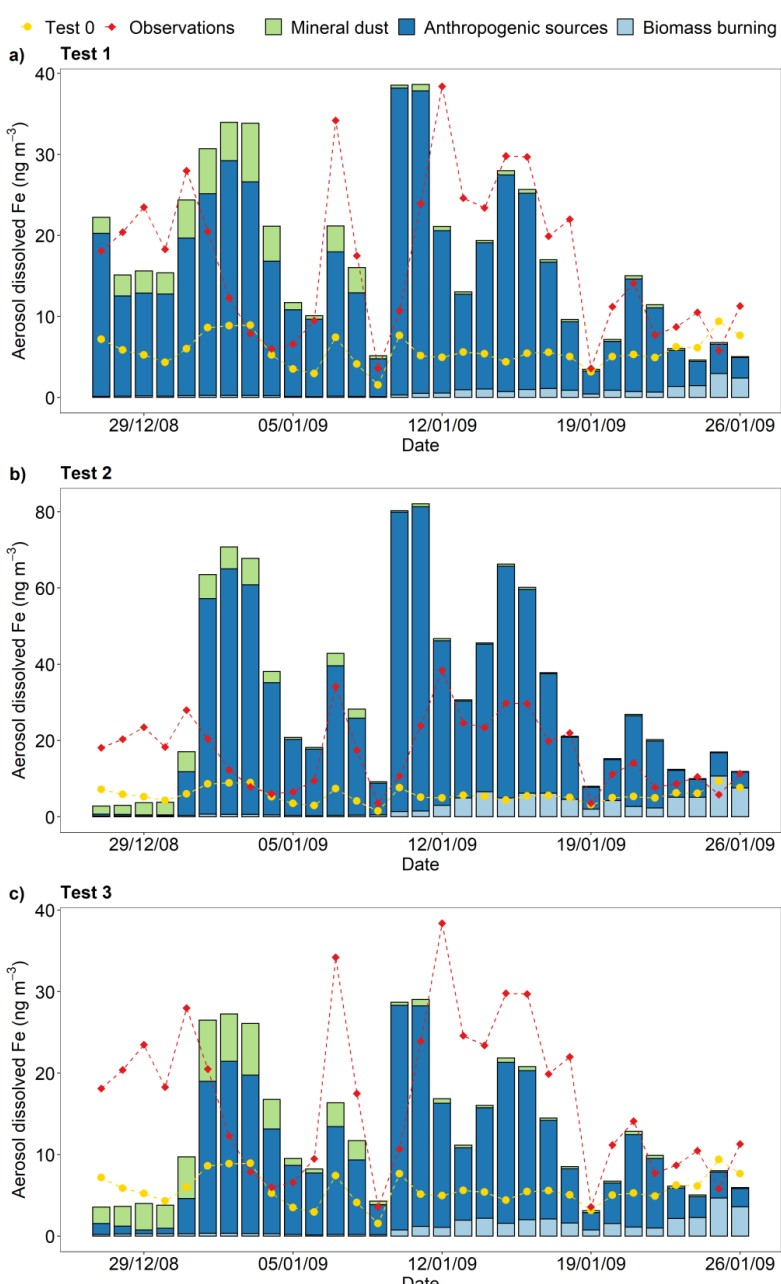

**Figure 8: Surface concentration of dissolved Fe in PM$_{2.5}$ aerosol particles over the Bay of Bengal from 27 December 2008 to 26 January 2009. Observations are from Bikkina et al. (2020) (red filled diamonds). Aerosol dissolved Fe was calculated along the cruise tracks using the IMPACT model. In Test 0 (yellow filled circles), we ran the model without upgrades in the Fe dissolution scheme (Ito et al., 2021a), and applying the proton-promoted, oxalate-promoted and photoinduced dissolution schemes for combustion aerosols. The contribution of mineral dust sources, anthropogenic sources and biomass burning is shown for Test 1-3. The proton + oxalate dissolution scheme (Table 1) was applied in Test 1 and 3, while proton-promoted dissolution is used for Test 2. We adopted the mineral-specific inventory for anthropogenic Fe emissions (Rathod et al., 2020) in Test 1 and 2. In Test 3, the Fe speciation of Krakow ash was used for all combustion sources.**



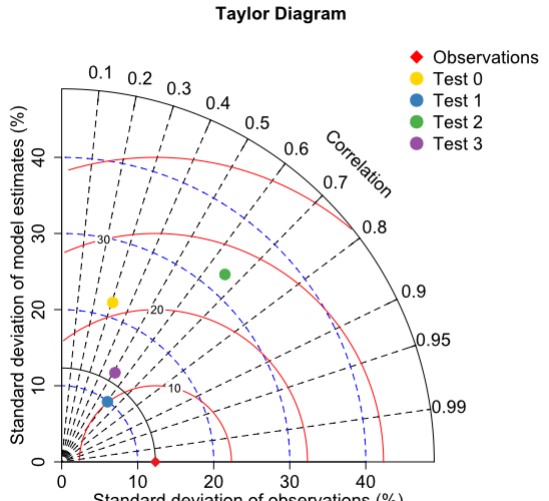


**Figure 9: Comparison between observations and model estimates of Fe solubility in PM₂.₅ aerosol particles over the Bay of Bengal**
**from 27 December 2008 to 26 January 2009. Observations are from Bikkina et al. (2020). Aerosol Fe solubility was calculated along**
**the cruise tracks using the IMPACT model. The Taylor diagram summarizes the statistics for the comparison between observations**
**of aerosol Fe solubility and the different simulations (Test 0-3). The dashed curves in blue indicate the standard deviation values.**
**The curves in red denote the root-mean-squared difference between the observational data and the model predictions. The dashed**
**lines in black represent the correlation coefficients.**

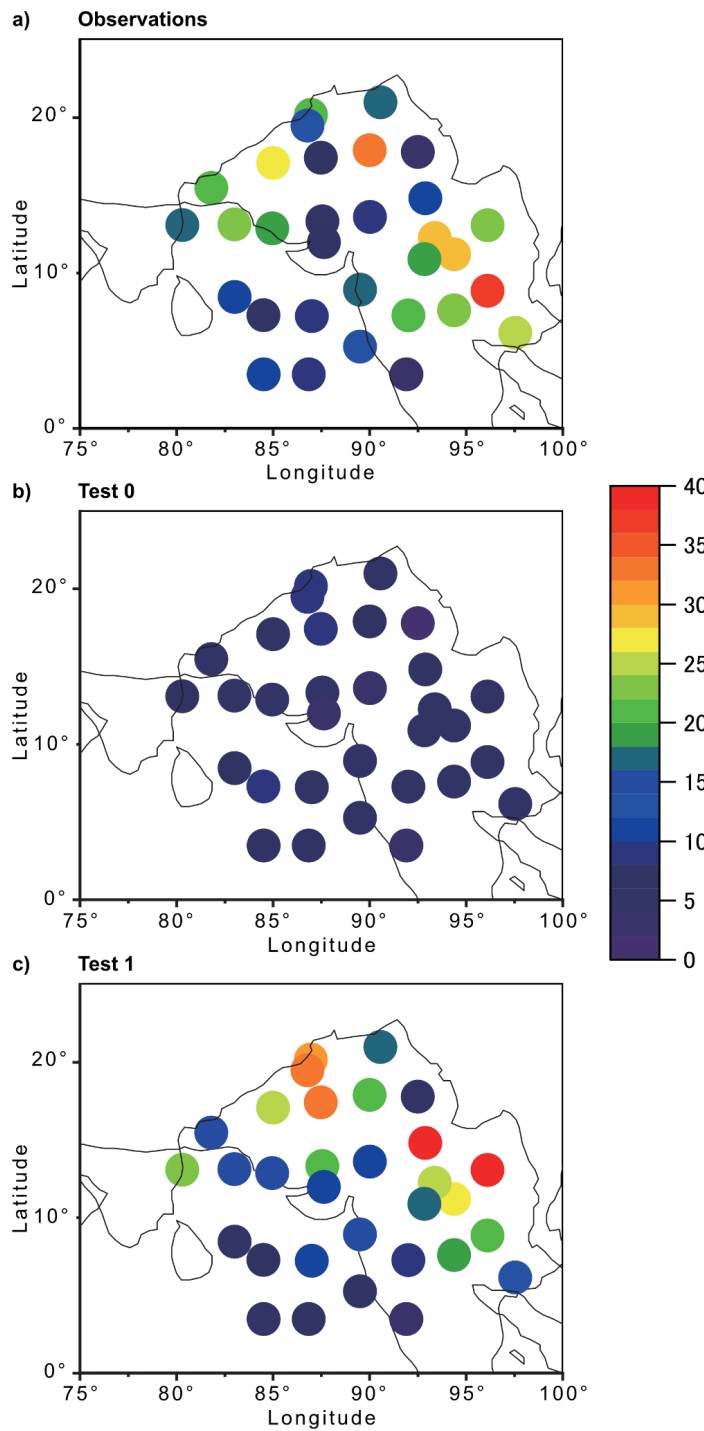


**Figure 10: Surface concentration of dissolved Fe in PM$_{2.5}$ aerosol particles over the Bay of Bengal from 27 December 2008 to 26 January 2009. a) Observations from Bikkina et al. (2020). b-c) Aerosol dissolved Fe calculated along the cruise tracks using the IMPACT model. In Test 0, we ran the model without upgrades in the Fe dissolution scheme (Ito et al., 2021a) and applying the proton-promoted, oxalate-promoted and photoinduced dissolution schemes for combustion aerosols Table S3 (Ito, 2015). The proton + oxalate dissolution scheme (Table 1) was applied in Test 1 and we adopted the mineral-specific inventory for anthropogenic Fe emissions (Rathod et al., 2020).**





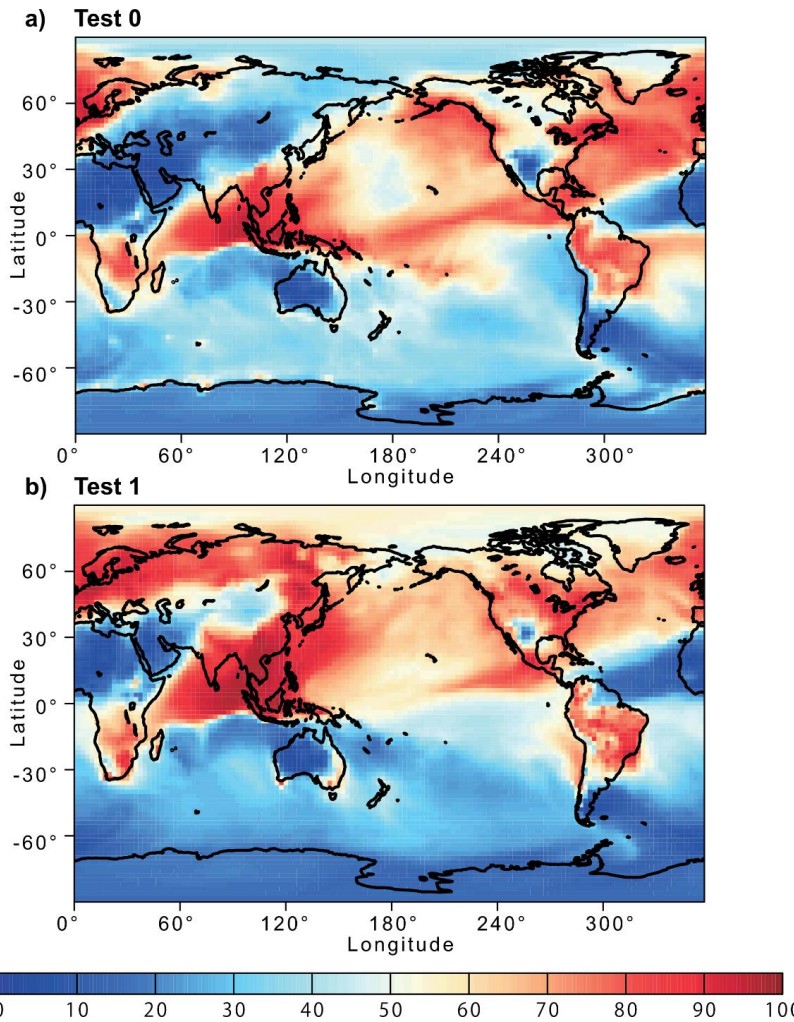

649

**Figure 11: Proportion (%) of pyrogenic dissolved Fe in aerosol dissolved Fe concentration near the surface from a) Test 0 and b) Test 1 for December 2008 and January 2009. In Test 0, we ran the model without upgrades in the Fe dissolution scheme (Ito et al., 2021a) and applying the proton-promoted, oxalate-promoted and photoinduced dissolution schemes for combustion aerosols Table S3 (Ito, 2015). The proton + oxalate dissolution scheme (Table 1) was applied in Test 1 and we adopted the mineral-specific inventory for anthropogenic Fe emissions (Rathod et al., 2020).**


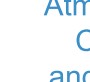 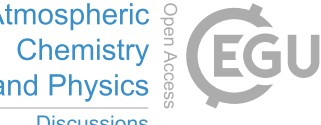

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
