# Peer review of "Iron from coal combustion particles dissolves much faster than mineral dust under simulated atmospheric acid conditions"

_Atmospheric Chemistry and Physics, 2021_

## Referee Comment (RC1)

[revised manuscript text omitted]

**Commented [RS(-S8]:** Change to bio-accessible here and in subsequent cases

**Commented [RS(-S9]:** Change to, 'The Fe transported in the atmosphere…'

**Commented [RS(-S10]:** Add an observational reference here to support this statement

**Commented [RS(-S11]:** Change this to < 1% and add refs Sholkovitz et al., 2009; 2012. Schroth et al., 2009, isn't the best ref here as they weren't comparing multiple dust samples.

**Commented [RS(-S12]:** Change to, '…processes occurring during atmospheric transport' as this would cover a wider range of mechanisms.
It would also be worth noting that although Fe solubility is low in mineral dust close to source regions, the shear volume of material deposited results in (relatively) high concentrations of soluble Fe.

**Commented [RS(-S13]:** Add metal smelting as a source

**Commented [RS(-S14]:** Add ref

**Commented [RS(-S15]:** Varies
I'm not clear what sources you mean here. Do you mean all sources or just the pyrogenic sources? You also need to make it clearer what the fractional solubility of oil and BB is higher than.

**Commented [RS(-S16]:** produced

**Commented [RS(-S17]:** principal

**Commented [RS(-S18]:** Fe-bearing particles

**Commented [RS(-S19]:** Best to change spelling to sulfur throughout as this is the IUPAC convention

[revised manuscript text omitted]

> **Commented [RS(-S23]:** Have you done recovery experiments to investigate microwave digestion using HNO3 only compared to a mixture of HNO3 and HF. It is usually necessary to include some HF for full digestion of mineral dust samples. This might be less of an issue CFAs but could potentially have resulted in incomplete digestion of the Libyan soil and ATD.

> **Commented [RS(-S24]:** This is useful information but more explanation is needed here. You haven't mentioned ATD before. Explain what it is and why you used it. Could you also include the relative proportions of each Fe mineral phase in ATD in the SI? This data is of interest as more people start to use ATD as an SRM

[revised manuscript text omitted]

**Commented [RS(-S26)]:** Write Libyan soil in brackets

**Commented [RS(-S27)]:** Than the Libyan dust end member

**Commented [RS(-S28)]:** Clarify if this the Libyan soil

**Commented [RS(-S29)]:** Slightly confusing labelling as Western Sahara is a country. If you can't change the label on the spectral plot, can you address this in the caption?

**Commented [RS(-S30)]:** I'm not seeing the double peaks for the Icelandic and Saharan dusts in the pre-edge region. Why do you think there are more similarities with the Icelandic dust? Also sourced from high temp processes or a coincidence?
Move Fig s2 into main text – you could have a Fig 5a and b c Author(s) 2021. CC BY 4.0
License.

[Figure]

[Figure]

the pre-edge region at around 7113.9 and 7115.2 eV, and a main peak in the edge region at around 7133.3 eV (Baldo et al.,
2020).

**4 Fe simulation from the IMPACT model**

**4.1 Fe dissolution scheme**

Based on the laboratory experiments carried out on the CFA samples, we implemented a 3-step dissolution scheme for proton-
promoted and oxalate-promoted Fe dissolution (Table 1). The Fe dissolution kinetics was described as follows (Ito, 2015):

$\sum_i RFe_i = k_i(pH, T) \times a(H^+)^{mi} \times f_i$                    (1)

where $RFe_i$ is the dissolution rate of individual mineral i, $k_i$ is the rate constant (moles Fe g$^{-1}$ s$^{-1}$), $a(H^+)$ is the H$^+$ activity in
solution, $m_i$ represents the empirical reaction order for protons. The function $f_i$ ($0 \leq f_i \leq 1$) accounts for the suppression of
mineral dissolution by competition for oxalate between surface Fe and dissolved Fe (Ito, 2015):

$f_i = 0.17 \times \ln([lig] \times [Fe]^{-1})_i + 0.63$                    (2)

in which, [Fe] is the molar concentration (mol L$^{-1}$) of Fe$^{3+}$ dissolved in solution, and [lig] is the molar concentration of ligand
(e.g., oxalate). $f_i$ was set to 1 for the proton-promoted dissolution.

The scheme assumes 3 rate constants "fast", "intermediate" and "slow" for the proton-promoted, and the proton + oxalate-
promoted dissolution (Table 1). These were obtained by fitting the parameters to our measurements for Krakow ash in H$_2$SO$_4$
and (NH$_4$)$_2$SO$_4$ at pH 2-3, with and without oxalate (Experiments 2 and 3 in Table S1), which are shown in Fig. 6. The fast rate
constant represents highly reactive Fe species such as amorphous Fe oxyhydroxides, Fe carbonates and Fe sulphates. The
intermediate rate constant can be applied to nano-particulate Fe oxides, while more stable phases including for example Fe
aluminosilicate and crystalline Fe oxides have generally slower rate (Ito and Shi, 2016; Shi et al., 2011a; Shi et al., 2011b; Shi
et al., 2015). Similarly, we predicted the dissolution kinetics of Aberthaw ash and Shandong ash (Figs. S3-S5). The dissolution
kinetic of Krakow ash was calculated based also on the experimental results at pH 1.0, which is shown in Fig. S6 in comparison
with kinetics predicted at pH 2.0 and pH 2.9 conditions.

**Commented [RS(-S31]:** The model is struggling to capture the initial rapid dissolution relevant to in-cloud processing. Why do you think this is?

**Commented [RS(-S32]:** kinetics

[revised manuscript text omitted]

**Commented [RS(-S37]:** A greater

**Commented [RS(-S38]:** rates

**Commented [RS(-S39]:** Fu et al used a different leach technique. They didn't include any oxalate. This different leach media will almost certainly impact dissolution kinetics and fractional solubility. In addition, you have already noted that the different CFAs in your study have different kinetics.

I suggest changing this to 'The Fe in our CFA samples initially dissolved faster than those used by Fu et al. (2012). This could be due to differences in the Fe speciation of the CFA samples in the two studies and/or the different leach media used.' – or words to that effect

**Commented [RS(-S40]:** In Chen et al's Fig. 4, it was 6-25% at pH 2 – this was the leach closest (but not the same) to your one. It was 21-70% after three pH cycles which is a very different leaching scheme to your one. The more appropriate comparison is with the data they show in Fig. 4 BUT you must note that your study and those of Fu and Chen all use quite different leach solutions and durations as some of the variability will result form these different conditions.

**Commented [RS(-S41]:** Note that this is in contrast to Chen et al's findings

**Commented [RS(-S42]:** Could also be c Author(s) 2021. CC BY 4.0
License.

[revised manuscript text omitted]

**Commented [RS(-S43]:** locations

**Commented [RS(-S44]:** this is a contradiction – estimate with more accuracy perhaps?

**Commented [RS(-S45]:** 'More'
More work to determine the mineralogy of Fe in 'natural' and processed CFAs, in combination with solubility experiments, in order to investigate the links between Fe solubility and Fe speciation/mineralogy is needed

**Commented [RS(-S46]:** I really think that the plots of your data that are currently in the SI should be moved into the main manuscript, as you refer to them quite extensively and it is much easier to follow if you don't have to keep flicking between the SI and manuscript. These plots could be consolidated, e.g. S8 and S9 could become one plot. Similarly, S3-5 could also become one plot. The tables should stay in the SI.

c Author(s) 2021. CC BY 4.0
License.

[Figure]

At pH 2.9, the proton + oxalate-promoted dissolution was higher than the proton + oxalate-promoted dissolution (Fig. S9d), with RFe$_{(oxalate)}$ > 0 (Eq. 5).

Moreover, the new 3-step dissolution scheme better captured the initial fast dissolution of CFA (Figs. 2-3) which was also observed in previous research (Borgatta et al., 2016; Chen et al., 2012; Chen and Grassian, 2013; Fu et al., 2012; Kim et al.,

2020) (except for the certified CFA 2689 in Chen et al. (2012) which showed increasing dissolution rates over the duration of the experiment). Furthermore, the new scheme enabled us to account for the different Fe speciation determined in the CFA

samples, which could be a key factor contributing to the different Fe dissolution behaviour observed in the present study and in literature (Borgatta et al., 2016; Chen et al., 2012; Chen and Grassian, 2013; Fu et al., 2012; Kim et al., 2020). In Figs. S3-

S5, the dissolution kinetics of Aberthaw and Shandong ash calculated using the dissolution rates in Table 1 and the Fe phases determined in the samples showed a good agreement with measurements.

**5.2 Comparison with mineral dust**

High ionic strength also impacted the dissolution rates of the Saharan dust sample at low pH (Fig. S10). At around pH 2

conditions, the proton-promoted Fe dissolution of Libya dust was enhanced by ~40% after the addition of (NH$_4$)$_2$SO$_4$. At around pH 2 and with 0.01 M H$_2$C$_2$O$_4$, the Fe solubility of Libya dust decreased by ~30% in the presence of (NH$_4$)$_2$SO$_4$. Overall, the

Fe solubility of Libya dust was lower compared to that observed in the CFA samples. After 168 hour-leaching at pH 2.1 with

1 M (NH$_4$)$_2$SO$_4$, the Fe solubility of Libya dust was 7.2% (Fig. S10), which was from around 3 to 7 times lower compared to that of the CFA samples (Fig. 1). At around pH 2 conditions in the presence of oxalate and high (NH$_4$)$_2$SO$_4$ concentration, the

Fe solubility of Libya dust rose to ~13.6% (Fig. S10), which is still 4 times lower than that of Krakow ash and around 1.5 lower than Aberthaw and Shandong ash (Fig. 2). The Fe solubilities of Libya dust observed in this study are comparable with those of the Tibesti dust in Ito and Shi (2016) at similar experimental conditions.

The enhanced Fe solubility in CFA compared to mineral dust could be primarily related to the different Fe speciation (Figs. 5

and S2). CFA contained more highly reactive Fe and magnetite but less hematite and goethite than mineral dust.

Although mineral dust is the largest contribution to aerosol Fe while CFA accounts for only a few percent, atmospheric processing of CFA may result in a larger than expected contribution of bioavailable Fe deposited to the surface ocean. It is thus important to quantify the amount and nature of CFA in atmospheric particles.

**5.3 Comparison of modelled Fe solubility with field measurements**

The model results obtained using the emission inventory from Rathod et al. (2020) and the new dissolution scheme for the proton + oxalate-promoted dissolution (Table 1) in Test 1 and Test 3 provided a better estimate of dissolved Fe over the Bay of Bengal than the other tests (Figs. 8, 9, and S7). At the same time, the new model improved the agreement of aerosol Fe solubility from Test 0 (70% ± 4%) to Test 1 (44% ± 3%) and Test 3 (48% ± 1%) with the field data (25% ± 3%) but still overestimated it after 22 January 2009, when open biomass burning sources become dominant (Fig. 8). This could be due to the unrepresentative Fe speciation used in Test 1 and Test 3 for biomass burning over the Bay of Bengal. To reduce the uncertainty in model predictions, emission inventories could be improved through a comprehensive characterization of Fe species in combustion aerosol particles.

**Commented [RS(-S47]:** Libyan rather than Libya. Change Saharan to Libyan to save confusion

**Commented [RS(-S48]:** I would assume that most people don't know where Tibesti is. I certainly didn't! State that the sample is also from a Saharan dust source region.

[revised manuscript text omitted]

---

## Referee Comment (RC2)

c Author(s) 2021. CC BY 4.0 License.

[Figure]

**1 Iron from coal combustion particles dissolves much faster than**
**2 mineral dust under simulated atmospheric acid conditions**

Clarissa Baldo[1], Akinori Ito[2], Michael D. Krom[3,4], Weijun Li[5], Tim Jones[6], Nick Drake[7], Konstantin
Ignatyev[8], Nicholas Davidson[1], Zongbo Shi[1]

[1]School of Geography Earth and Environmental Sciences, University of
Birmingham, Birmingham, United Kingdom

[2]Yokohama Institute for Earth Sciences, JAMSTEC, Yokohama, Kanagawa 236-
0001, Japan

[3]Morris Kahn Marine Station, Charney School of Marine Sciences, University of
Haifa, Haifa, Israel

[4]School of Earth and Environment, University of Leeds, Leeds, United Kingdom

[5]Department of Atmospheric Sciences, School of Earth Sciences, Zhejiang
University, Hangzhou 310027, China

[6]School of Earth and Environmental Sciences, Cardiff University, Cardiff, United
Kingdom

[7]Department of Geography, King's College London, London, United Kingdom

[8]Diamond Light Source, Didcot, Oxfordshire, United Kingdom

*Correspondence to*: Zongbo Shi (z.shi@bham.ac.uk); Akinori Ito
(akinori@jamstec.go.jp)

**Abstract.** Mineral dust is the largest source of total aerosol iron (Fe) loading over
the offshore global ocean, but acidic processing of coal fly ash (CFA) in the
atmosphere may result in a disproportionally higher contribution of bioavailable Fe.
Here, we determined the Fe speciation and dissolution kinetics of CFA from
Aberthaw (United Kingdom), Krakow (Poland), and Shandong (China) in solutions
which simulate atmospheric acidic processing. In CFA-$PM_{10}$ fractions, 8%-21.5%
of the total Fe was as hematite and goethite (dithionite extracted Fe), 2%-6.5 % as
amorphous Fe (ascorbate extracted Fe), while magnetite (oxalate extracted
Fe) varied from 3%-22%. The remaining 50%-87 % of Fe was associated with
aluminosilicates. High concentration of ammonium sulphate (($NH_4$)$_2SO_4$), often
found in wet aerosols, increased Fe solubility of CFA up to 7 times at pH 2-3. Our

Commented [MP1]: This hasn't been shown (to this extent) by any study. In order to have such impact CFA emissions would have to be both highly soluble (they are) and of very large magnitude (not shown to date). Maybe just pace own the sentence by chosing another word.

Commented [MP2]: Associated figure says "other" I cannot recall any data showing the remaining fraction is aluminosilicate. If no proof maybe correct to "remaining 50%-87 % of Fe was thought to be associated with ..." or " remaining 50%-87 % of Fe may be comprised in …"

c Author(s) 2021. CC BY 4.0 License.

[Figure]

results showed a large variability in the effects of oxalate on the Fe dissolution rates at pH 2, from no impact in Shandong ash to doubled dissolution in Krakow ash.

However, this enhancement was suppressed in the presence of high concentration of

$(NH_4)_2SO_4$. Dissolution of highly reactive Fe was insufficient to explain the high Fe solubility at low pH in CFA, and the modelled dissolution kinetics suggests that other Fe phases such as magnetite may also dissolve rapidly under acidic conditions. Overall, Fe in CFA dissolved up to 7 times faster than in Saharan dust samples at pH 2. Based on these laboratory data, we developed a new scheme for the proton- and oxalate- promoted Fe dissolution of CFA, which was implemented into the global atmospheric chemical transport model IMPACT. The revised model showed a better agreement with observations of  of dissolved Fe concentration in aerosol particles over the Bay of Bengal, due to the rapid Fe release at the initial stage at highly acidic conditions. The improved model also enabled us to predict sensitivity to a more dynamic range of pH changes, particularly between anthropogenic combustion and biomass burning aerosols.

**1 Introduction**

The availability of iron (Fe) limits primary productivity in high-nutrient low- chlorophyll (HNLC) regions of the global ocean including the subarctic North

Pacific, the East Equatorial Pacific and the Southern Ocean (Boyd et al., 2007;

Martin, 1990). In other regions of the global ocean such as the subtropical North

Atlantic, the Fe input may affect primary productivity by stimulating nitrogen fixation (Mills et al., 2004; Moore et al., 2006). These areas are particularly sensitive to changes in the supply of bioavailable Fe. Atmospheric aerosols are an important source of soluble (and, thus potentially bioavailable) Fe to the offshore global ocean. The deposition of bioavailable Fe to the ocean can alter biogeochemical cycles and increase the carbon uptake, consequently affecting the climate (e.g., Jickells and Moore, 2015; Jickells et al., 2005; Kanakidou et al., 2018;

Commented [MP3]: Either because of the pH or because of differences in the samples…? I'd delete this sentence this is not the main finding

Commented [MP4]: No word on oxalate addition?

Commented [MP5]: Lybia (or just 'our reference dust sample')

Commented [MP6]: I do not think the model focuses on surface Fe dissolution only. Likely it accounts for Fe dissolution from one aerosol as a whole (although it is thought that surface Fe will dissolve first).

Commented [MP7]: Community now tries to use "bioaccessible" Fe as the genuine bioavailability is still to prove.

[Figure]

Mahowald et al., 2010; Shi et al., 2012). In general, bioavailable Fe consists of aerosol dissolved Fe, and Fe-nanoparticles which can be present in the original particulate matter and/or formed during atmospheric transport as a result of cycling into and out of clouds (Shi et al., 2009). It is in addition possible that other more refractory forms of Fe could be solubilised in the surface waters by zooplankton (Schlosser et al., 2018) or the microbial community (Rubin et al., 2011).

Atmospheric Fe is largely derived from lithogenic sources, which contribute around

95% of the total Fe in suspended particles (e.g., Myriokefalitakis et al., 2018) and hence most studies concentrate on atmospheric processing of mineral dust (e.g.,

Cwiertny et al., 2008; Fu et al., 2010; Ito and Shi, 2016; Shi et al., 2011a; Shi et al.,

2015). Mineral dust has low Fe solubility (dissolved Fe/ total Fe) near the source regions, generally below 0.5% (e.g., Schroth et al., 2009; Shi et al., 2011c), increasing somewhat as a result of atmospheric processing (e.g., Baker et al., 2021;

Baker et al., 2020). Other sources of bioavailable Fe to the ocean are from combustion sources such as biomass burning, coal combustion and oil combustion (e.g., shipping emissions) (e.g., Ito et al., 2018; Rathod et al., 2020). Although these sources are only a small fraction of the total Fe in atmospheric particulates, the Fe solubility of pyrogenic sources can be 1–2 orders of magnitude higher than in mineral dust, and thus can be important in promoting carbon uptake. However, the

Fe solubility of these sources vary considerably depending on the particular sources with higher values observed for oil combustion and biomass burning (Ito et al.,

2021b and references therein).

Wang et al. (2015) estimated that coal combustion produces around ~0.9 Tg yr$^{-1}$ of atmospheric Fe (on average for 1960– 2007), contributing up to ~86% of the total anthropogenic Fe emissions. A more recent study, which has included metal

Commented [MP8]: Rephrase maybe 'most studies so far have concentrated on ..'

Commented [MP9]: I would say <1% is more accurate according to published literature

I'd suggest citing
Sholkovitz et al 2012
http://dx.doi.org/10.1016/j.gca.2012.04.022

Sholkovitz et al 2009
10.1016/j.gca.2009.04.029

Commented [MP10]: Reference needed

Commented [MP11]: May be worth citing a lab-based or field-based study as well here.

Commented [MP12]: Emitted?

Commented [MP13]: Rephrase maybe '…of Fe into the atmosphere'

c Author(s) 2021. CC BY 4.0 License.

[revised manuscript text omitted]

Fe dissolution behaviour in the presence of oxalate at pH 1.0.

**Commented [MP32]:** On Fig 4 it looks like the addition of $(NH_4)_2SO_4$ does slightly (yet noticeably) diminish the Fe dissolution. Or is that another factor coming into play?

**3.2 Fe speciation**

The Fe phases in the CFA samples determined through sequential extractions are shown in Fig. 5. The Fe speciation in the Saharan dust sample is added for

**Commented [MP33]:** If this is the Libya dust, please keep the name consistent throughout comparison. Krakow ash had a total Fe (FeT) content of 5.2%, while FeT in

Aberthaw and Shandong ash was 3.1% and 1.6% respectively. Amorphous Fe (FeA/FeT) was 6.5% in Krakow ash, 2% in Aberthaw ash, and 4.6% in Shandong ash. The CFA samples showed very different dithionite Fe (FeD/FeT) content,

21.5% in Krakow ash, 8% in Aberthaw ash and 14.8% in Shandong ash. The content of magnetite (FeM/FeT) was considerably higher in Krakow ash (22.4%)

compared to Aberthaw (2.9%) and Shandong (4.5%) ash. About 50 %–87 % of Fe was contained in other phases most likely in aluminosilicates. Overall, CFA had (c) Author(s) 2021. CC BY 4.0 License.

[Figure]

[Figure]

more magnetite and highly reactive amorphous Fe and less dithionite Fe than Libya dust.

In Fig. S2, the Fe K-edge XANES spectra of Krakow and Aberthaw ash showed a single peak in the pre-edge region at around

7114.3 eV and 7114.6 eV, respectively. In the edge region, Aberthaw ash showed a broad peak at around 7132.2 eV, while the peak of Krakow ash was slightly shifted to 7132.9 eV and narrower. The pre-edge peak at around 7115.4 suggest that Fe was mainly as Fe(III). The spectral features of Aberthaw and Krakow ash are different from those of the hematite, magnetite and illite standards suggesting that the glass fraction was dominant and controlled their spectral characteristics, which is consistent with the results of the Fe sequential extractions. The XANES Fe K-edge spectra of the CFA samples have some common features with those of Icelandic dust, but differs from northern African dust (Fig. S2). Aluminium silicate glass is also dominant in Icelandic dust (Baldo et al., 2020). In the pre-edge region,

Icelandic dust (sample MIR 45 in Fig. S2) showed a main peak at around

7114.4 eV and a second less intense peak at around 7112.7 eV, while a broad peak was observed at around 7131.9 eV in the edge region (Baldo et al., 2020). Northern

African dust (western Sahara in Fig. S2) showed a distinct double peak in the pre- edge region at around 7113.9 and 7115.2 eV, and a main peak in the edge region at around 7133.3 eV (Baldo et al., 2020).

**4 Fe simulation from the IMPACT model**

**4.1 Fe dissolution scheme**

Based on the laboratory experiments carried out on the CFA samples, we implemented a 3-step dissolution scheme for protonpromoted and oxalate-promoted

Fe dissolution (Table 1). The Fe dissolution kinetics was described as follows (Ito,

2015):

**Commented [MP34]:** Please develop slightly the reasoning here

**Commented [MP35]:** This is a funny parallel between CFA and dust, why would similarities be expected? Is that in order to better understand the composition (mineralogy) of the CFA? Is the exact composition of the Icelandic dust known? Is it dust or aerosols collected at a random location in Iceland? Maybe a word to understand why the parallel between CFA and dust is made would help.

**Commented [MP36]:** Libya dust?

**Commented [MP37]:** Libya dust? Please keep wording consistent

**Commented [MP38]:** I cannot see the double peak in the pre-edge region though I do not have a trained eye on this kind of graph. Maybe it isn't that distinct?

$\sum_i RFe_i = k_i(pH, T) \times a(H^+)^{mi} \times f_i$

        (1)

[revised manuscript text omitted]

**Commented [MP42]:** three

**Commented [MP43]:** which are mostly

**Commented [MP44]:** do you mean total surface aerosol Fe? Vs surface dissolved Fe?  The term aerosol Fe is confusing as dissolved Fe also refers to Fe which is part of aerosol particles

**Commented [MP45]:** How did measurements differentiate surface aerosol Fe to total aerosol Fe and surface dissolved aerosol Fe to total dissolvable aerosol Fe?
Are all the measurements really specific to the surface of aerosols or is dissolved Fe only assumed to originate from the surface of aerosol (what is the aerosol Fe component then?)?

Maybe better wording for each measurement is required for enhanced clarity.

**Commented [MP46]:** Total Fe in aerosols I assume?

**Commented [MP47]:** May I suggest that biomass burning is represented in orange or another colour as we cannot see it well as it currently is.
Also, maybe instead of / in addition to plotting date as x axis, the location could be stated/ indicated (from the Northern Bay of Bengal to the South).

c Author(s) 2021. CC BY 4.0 License.

[Figure]

studies indicating that the aerosol Fe concentrations over the North Bay of Bengal are influenced by emissions of dust and combustion sources from the Indo-Gangetic

Plain (Kumar et al., 2010), whereas combustion sources (e.g., biomass burning and fossil-fuel) from South-East Asia are dominant over the South Bay of Bengal (Kumar et al., 2010; Srinivas and Sarin, 2013). On the other hand, the model could not reproduce the peak in total Fe concentration (1.8% of Fe content in $PM_{2.5}$

sample) reported around 29 December 2008. The total Fe observed in $PM_{2.5}$ (613 ng

$m^{-3}$) is higher than that in $PM_{10}$ (430 ng $m^{-3}$) (Srinivas et al., 2012). This may be due to the measurement uncertainty including sample collection with two different high- volume samplers (Kumar et al., 2010).

The average dissolved Fe concentration in aerosols measured over the North Bay of

Bengal (16 ± 9 ng $m^{-3}$) is slightly lower than that over the South Bay of Bengal (18

± 10 ng $m^{-3}$) (Bikkina et al., 2020). The model prediction of dissolved Fe over the

North Bay of Bengal was 6 ± 2 ng $m^{-3}$ Fe in Test 0, 21 ± 10 ng $m^{-3}$ in Test 1, and

31 ± 28 ng $m^{-3}$ in Test 2, and 13 ± 10 ng $m^{-3}$ in Test 3. The aerosol dissolved Fe estimated over the South Bay of Bengal was 6 ± 1 ng $m^{-3}$ in Test 0, 15 ± 10 ng $m^{-3}$

in Test 1, 32 ± 22 ng $m^{-3}$ in Test 2, and 12 ± 7 ng $m^{-3}$ in Test 3. In Fig. 8, our model results show that the contribution of mineral dust to aerosol dissolved Fe was higher over the North Bay of Bengal (14% ± 6% in Test 1, 28% ± 34% in Test 2, and 33%

± 26% in Test 3) compared to the South Bay of Bengal (3% ± 1% in Test 1, 1% ±

1% in Test 2, and 3% ± 1% in Test 3). Overall, anthropogenic combustion sources were dominant over the Bay of Bengal accounting for 84% ± 12% in Test 1, 72% ±

29% in Test 2, and 69% ± 24% in Test 3 of the aerosol dissolved Fe. Moreover, after 22 January 2009, the contribution of open biomass burning sources increased up to 47% in Test 1, 64% in Test 2, and 60% in Test 3 (Fig. 8).

**Commented [MP48]:** This peak is quite large for reflecting measurement uncertainty isn't it?
Have you checked whether this point may be the result of a specific weather event (fire emission for example, such a peak may be visible from satellite observation) or may originate from a single sample contamination (in this case it should be discarded).

**Commented [MP49]:** dFe concentrations in aerosols seem to fluctuate a lot throughout the sampling transect. I would personally use median value rather than average for aerosols to avoir being biased by very low/high data points.
The lowest dissolved Fe data are towards the right-hand side of the graph which I understood as "southern Bay of Bengal" but with no spatial indication on the graph I may have mis-interpreted the data.

c Author(s) 2021. CC BY 4.0 License.

[Figure]

The aerosol Fe solubility measured over the South Bay of Bengal is higher than that over the North Bay of Bengal, respectively 32% ± 11% and 15% ± 7% (Bikkina et al., 2020), and model estimates showed a similar trend (Fig. S7). In Fig. S7, the calculated average Fe solubility over the North Bay of Bengal in Test 3 (18% ±

10%) was in good agreement with observations, while lower Fe solubility was estimated in Test 0 (8% ± 5%) and higher values were obtained for Test 1 (28% ±

8%). The aerosol Fe solubility over the South Bay of Bengal was better captured in

Test 1 (43% ± 4%) and Test 3 (39% ± 7%), whereas Test 0 showed higher variability (38% ± 22%). The proton-promoted dissolution scheme in Test 2

significantly overestimated the Fe solubility over the Bay of Bengal (Figs. 9 and

S7). The aerosol Fe solubility was largely overestimated in all scenarios after 22

January 2009, as open biomass burning sources become dominant (Fig. 8). The comparison between observations and model predictions of Fe solubility over the

Bay of Bengal is shown in Fig. 9. The agreement between measurements and model predictions was the best in Test 1 and Test 3. These exhibited good correlation with observations (R = 0.60 in Test 1 and R = 0.51 in Test 3), and the lowest centred root-mean-square (RMS) difference between the simulated and observed aerosol Fe solubilities (RMS = 16 in Test 1 and RMS = 14 in Test 3). In Test 0, the model estimates showed higher difference from observations (RMS = 22) and poor correlation (R = 0.30).

**5 Discussion**

**5.1 Dissolution behaviour of Fe in CFA**

In this study, the Fe dissolution kinetics of CFA samples from UK, Poland and

China was investigated under simulated atmospheric acidic conditions. A key parameter in both the atmosphere and the simulation experiments is the pH of the water interacting with the CFA particles. The lower the pH of the experimental solution the faster the dissolution and eventually the higher the amount of Fe

> **Commented [MP50]:** I would personally have chosen Fig S7 to display on the ms instead of Fig 9. Indeed, Fig S7 as it shows the performance of each test-run to reproduce the observational data. Fig 9 is great but a little complex to interpret and to assess the model performance from.

c Author(s) 2021. CC BY 4.0 License.

[Figure]

dissolved. Our results showed a strong pH dependence in low ionic strength conditions, with higher dissolution rate at lower pH. For example, reducing the solution pH from 2.7 to 2.1, the Fe solubility of Krakow ash increased by a factor of

4 (Fig. 1a) over the duration of the experiments, while the Fe solubility of Aberthaw and Shandong ash increased by 9-10 times from pH 2.9 to pH 2.2 (Figs. 1b-c). This enhancement is higher than that observed in studies conducted on mineral dust samples, which showed that one pH unit can lead to 3-4 times difference in dissolution rates (Ito and Shi, 2016; Shi et al., 2011a). Furthermore, Chen et al.

(2012) reported that the Fe solubility of the certified CFA 2689 only increased by

10% from pH 2 to pH 1, after 50 hours of dissolution in acidic media. The Fe solubility of CFA ($PM_{10}$ fractions) after 6 hours at pH 2 was 6%-10% for Aberthaw and Shandong ash respectively, and 28% for Krakow ash (Fig. 1). These values are higher than the Fe solubilities measured by Fu et al. (2012), who reported 2.9%-

4.2% Fe solubility in bulk CFA from three coal-fired power plants in China after

12-hour leaching at pH 2. This suggest that Fe in our CFA samples initially dissolved faster than those used in Fu et al. (2012). The Fe solubility after 72-hour leaching in $H_2SO_4$ at around pH 2 varied from around 12% and 17% (Aberthaw and

Shandong ash) to 34% (Krakow ash). These values are at the lower end of the range or below those reported in Chen et al. (2012), who measured a Fe solubility of

~20%-70% in certified CFA samples after accumulated acid dissolution of 72 hours at pH 2. These results suggest that there are considerable variabilities in the pH

dependent dissolution of Fe in CFA.

Our results showed that high ionic strength has a major impact on dissolution rates of CFA at low pH (i.e., pH 2-3). The Fe solubility of CFA increased by approximatively 20%-40% in the presence of 1 M $(NH_4)_2SO_4$ at around pH 2 over

**Commented [MP51]:** Different samples and leaching protocols and leaching solutions will also play a role in the differences highlighted here and above. 3 CFAs in this study already have different reactions to the same protocol, likely due to different mineralogy and different degree of combustion (this study has not addressed possible incomplete combustion from the sampled power plants)

**Commented [MP52]:** Please see comment above, I believe many factors come into play and this conclusion cannot be drawn without acknowledging these parameters.

**Commented [MP53]:** Here may also be a good time to mention other parameters that come into play apart from pH. Reminding the readers the all these studies are not strictly comparable to one another.

**Commented [MP54]:** Is that the percentage increase or the final Fe solubility. Maybe the "from x%…to x%" is less prone to misunderstanding.

[revised manuscript text omitted]

**Commented [MP59]:** Less soluble phases? I believe they are all chemically stable but show different reactivity toward the leaching media

intermediate pool were in good agreements with measurements (Figs. S3-S6). FeM

is likely to be primary magnetite but may contain a fraction of the more reactive aluminosilicate glass. Our model results suggest that magnetite in CFA particles may be more soluble than has been shown in Marcotte et al. (2020). It is possible that in real CFA samples the physicochemical properties of minerals, including for example crystal size, degree of crystallinity, cationic and anionic substitution in the lattice which influence the Fe dissolution behaviour (e.g., Schwertmann, 1991), are likely to be different from those of the reference minerals analysed in Marcotte et al.

(2020). In order to estimate in detail the relative contribution of different mineral phases present in CFAs to dissolved Fe, most detailed work would be needed to determine Fe mineral phases in pristine and processed CFA particles.

Finally, the modelled dissolution kinetics obtained using the new dissolution scheme for CFA (Table 1) showed better agreement with laboratory measurements than when using the original scheme (Ito, 2015) (Figs. S8 and S9). In Fig. S8, we compared the Fe dissolution kinetics of Krakow ash at around pH 2 and 3 with 1 M

$(NH_4)_2SO_4$ calculated using the proton-promoted dissolution scheme in Table 1

with the dissolution kinetics calculated at similar pH but using the proton-promoted dissolution scheme for combustion aerosols in Ito (2015) (Table S3). The dissolution scheme in Ito (2015) was based on laboratory measurements conducted at low ionic strength (Chen et al., 2012) and assumed a single Fe phase in combustion aerosol particles, while the new dissolution scheme considered the high ionic strength of aerosol water and assumed three rate constants, for fast, intermediate and slow kinetics of the different Fe phases present in CFA particles.

The Fe dissolution kinetics obtained using the new dissolution scheme showed a better agreement with measurements and was enhanced compared to the model estimates obtained using the original dissolution scheme (Ito, 2015) for low ionic strength conditions (Fig. S8). Figure S9 shows the Fe dissolution kinetics of

**Commented [MP60]:** What is a pristine CFA? Aren't CFAs always the result of a combustion process (therefore, 'processed' particles)? Not sure about the wording used.

**Commented [MP61]:** May be worth displaying in the manuscript c Author(s) 2021. CC BY 4.0 License.

[revised manuscript text omitted]

c Author(s) 2021. CC BY 4.0 License.

[Figure]

[Figure]

**Commented [MP69]:** BB could have been another colour (orange for example) for a better visual.

Figure 8: Surface concentration of dissolved Fe in PM$_{2.5}$ aerosol particles over the Bay of Bengal from 27 December 2008 to 26 January 2009. Observations are from Bikkina et al. (2020) (red filled diamonds). Aerosol dissolved Fe was calculated along the cruise tracks using the IMPACT model. In Test 0 (yellow filled circles), we ran the model without upgrades in the Fe dissolution scheme (Ito et al., 2021a), and applying the proton-promoted, oxalate-promoted and photoinduced dissolution schemes for combustion aerosols. The contribution of mineral dust sources, anthropogenic sources and biomass burning is shown for Test 1-3. The proton + oxalate dissolution scheme (Table 1) was applied in Test 1 and 3, while proton-promoted dissolution is used for Test 2. We adopted the mineral-specific inventory for anthropogenic Fe emissions (Rathod et al., 2020) in Test 1 and 2. In Test 3, the Fe speciation of Krakow ash was used for all combustion sources.

c Author(s) 2021. CC BY 4.0 License.

[Figure]

[Figure]

**Figure 9: Comparison between observations and model estimates of Fe solubility in PM$_{2.5}$ aerosol particles over the Bay of Bengal from 27 December 2008 to 26 January 2009. Observations are from Bikkina et al. (2020). Aerosol Fe solubility was calculated along the cruise tracks using the IMPACT model. The Taylor diagram summarizes the statistics for the comparison between observations of aerosol Fe solubility and the different simulations (Test 0-3). The dashed curves in blue indicate the standard deviation values. The curves in red denote the root-mean-squared difference between the observational data and the model predictions. The dashed lines in black represent the correlation coefficients.**

[Figure]

[Figure]

**Commented [MP70]:** Can the color scale be improved in the dark blue and purple shades?

**Figure 10: Surface concentration of dissolved Fe in PM$_{2.5}$ aerosol particles over the Bay of Bengal from 27 December 2008 to 26 January 2009. a) Observations from Bikkina et al. (2020). b-c) Aerosol dissolved Fe calculated along the cruise tracks using the IMPACT model. In Test 0, we ran the model without upgrades in the Fe dissolution scheme (Ito et al., 2021a) and applying the**

c Author(s) 2021. CC BY 4.0 License.

proton-promoted, oxalate-promoted and photoinduced dissolution schemes for combustion aerosols Table S3 (Ito, 2015). The
proton + oxalate dissolution scheme (Table 1) was applied in Test 1 and we adopted the mineral-specific inventory for
anthropogenic Fe emissions (Rathod et al., 2020).

[Figure]

**Figure 11: Proportion (%) of pyrogenic dissolved Fe in aerosol dissolved Fe concentration near the surface from a) Test 0 and**
**b) Test 1 for December 2008 and January 2009. In Test 0, we ran the model without upgrades in the Fe dissolution scheme (Ito**
**et al., 2021a) and applying the proton-promoted, oxalate-promoted and photoinduced dissolution schemes for combustion**
**aerosols Table S3 (Ito, 2015). The proton + oxalate dissolution scheme (Table 1) was applied in Test 1 and we adopted the**
**mineral-specific inventory for anthropogenic Fe emissions (Rathod et al., 2020).**

**Commented [MP71]:** Is that only the anthropogenic fraction of pyrogenic Fe? Maybe stick to CFA-Fe or anthropogenic-Fe if more accurate?

I would suggest to change the caption to 'Contribution of pyrogenic aerosols to the atmospheric dissolved Fe loading…'

**Commented [MP72]:** Maybe Figures 10 and 11 could go to SI as they are only very briefly discussed

[revised manuscript text omitted]

---

## Author Comment (AC1)

The description and discussions of the model results in sections 2.5, 4.2, 5.3 have been updated. The new model data are reported in Tables S4 and S5. Figures 8, 9, 11, and 12 of the revised manuscript show new model results. The model data available at https://doi.org/10.25500/edata.bham.00000702 have been also updated.

**Referee 1 – Dr Rachel Shelley**

General Response: We thank Dr Shelley for providing constructive comments. The comments have been addressed point by point below.

**Comment 1: Soluble or labile would be better choices here**

Response: Page 1, lines 14-16 have been updated as follows:

"Mineral dust is the largest source of aerosol iron (Fe) to the offshore global ocean, but acidic processing of coal fly ash (CFA) in the atmosphere could be an important source of soluble aerosol Fe".

**Comment 2: Given in the main manuscript.**

Response: The reviewer comment has been implemented in the manuscript (Page 1, line 25).

**Comment 3: Given in the main manuscript.**

Response: The reviewer comment has been implemented in the manuscript (Page 1, line 25).

**Comment 4: Given in the main manuscript.**

Response: The reviewer comment has been implemented in the manuscript (Page 1, line 27).

**Comment 5: Given in the main manuscript.**

Response: The reviewer comment has been implemented in the manuscript (Page 1, line 28).

**Comment 6: Given in the main manuscript.**

Response: "Surface concentrations of dissolve Fe" (Page 1, line 31) has been replaced with Fe solubility.

**Comment 7: replace with '…due to the initial rapid release of Fe at low pH.**

Response: Page 1, lines 30-34 have been updated as follows:

"The revised model showed a better agreement with observations of Fe solubility in aerosol particles over the Bay of Bengal, due to the initial rapid release of Fe and the suppression of the oxalate-promoted dissolution at low pH".

**Comment 8: Change to bio-accessible here and in subsequent cases**

Response: The term "Bioavailable Fe" has been replaced with "Bio-accessible Fe" throughout the manuscript.

**Comments 9-15: (9) Change to, 'The Fe transported in the atmosphere…'. (10) Add an observational reference here to support this statement. (11) Change this to < 1% and add refs Sholkovitz et al., 2009; 2012. Schroth et al., 2009, isn't the best ref here as they weren't comparing multiple dust samples. (12) Change to, '…processes occurring during atmospheric transport' as this would cover a wider range of mechanisms. It would also be worth noting that although Fe solubility is low in mineral dust close to source regions, the shear volume of material deposited results in (relatively) high concentrations of soluble Fe. (13) Add metal smelting as a source. (14) Add ref. (15) Varies. I'm not clear what sources you mean here. Do you mean all sources or just the pyrogenic sources? You also need to make it clearer what the fractional solubility of oil and BB is higher than.**

Response: Page 2, lines 49-60 have been updated as follows:

"The Fe transported in the atmosphere is largely derived from lithogenic sources, which contribute around 95% of the total Fe in suspended particles (e.g.,Shelley et al., 2018) and most studies so far have concentrated on atmospheric processing of mineral dust (e.g., Cwiertny et al., 2008; Fu et al., 2010; Ito and Shi, 2016; Shi et al., 2011a; Shi et al., 2015). Mineral dust has low Fe solubility (dissolved Fe/ total Fe) near the source regions, generally below 1% (e.g., Shi et al., 2011c; Sholkovitz et al., 2009; Sholkovitz et al., 2012), increasing somewhat as a result of processes occurring during atmospheric transport (e.g., Baker et al., 2021; Baker et al., 2020). Other sources of bio-accessible Fe to the ocean are from combustion sources such as biomass burning, coal combustion, oil combustion, and metal smelting (e.g., Ito et al., 2018; Rathod et al., 2020). Although these sources are only a small fraction of the total Fe in atmospheric particulates, the Fe solubility of pyrogenic sources can be 1–2 orders of magnitude higher than in mineral dust (Ito et al., 2021b and references therein), and thus can be important in promoting carbon uptake. However the Fe solubility of pyrogenic sources varies considerably depending on the particular sources with higher values observed for oil combustion and biomass burning than coal combustion sources (Ito et al., 2021b and references therein)"

**Comments 16-18: Given in the main manuscript.**

Response: The reviewer comments have been implemented in the manuscript (Page 2, lines 61, 65, and 70).

**Comment 19: Best to change spelling to sulfur throughout as this is the IUPAC convention**

Response: The reviewer comment has been implemented throughout the manuscript.

**Comment 20: Given in the main manuscript.**

Response: The reviewer comments have been implemented in the manuscript (Page 3, line 89).

**Comment 21: Given in the main manuscript.**

Response: The reviewer comments have been implemented in the manuscript (Page 3, line 101).

**Comment 22: As soil isn't mineral dust, you will need to justify its use, which is that you are using it as an analogue for a Saharan mineral dust end member in the same way that you are using the CFA as an end member for CFA dust in the atmosphere. Later in the text you sometimes refer to Libyan dust, sometimes Saharan dust and in the XANES spectral plot, as western Saharan dust. For consistency, and to avoid confusion, best to stick to one.**

Response: Libyan dust precursor is now specified throughout the manuscript.

In addition, Page 4, lines 136-139 have been updated as follows: "A soil sample from Libya (Soil 5, 32.29237N/22.30437E) was dry sieved to 63 µm and used as an analogue for a Saharan mineral dust precursor to make a comparison between CFA and mineral dust".

**Comment 23: Have you done recovery experiments to investigate microwave digestion using HNO3 only compared to a mixture of HNO3 and HF. It is usually necessary to include some HF for full digestion of mineral dust samples. This might be less of an issue CFAs but could potentially have resulted in incomplete digestion of the Libyan soil and ATD.**

Response: For the analysis of the total Fe content, we only used the microwave digestion in concentrated $HNO_3$. The recovery of Fe was around 89% which was assessed using a standard reference material for urban particulate matter (NIST SRM 1648A). We did not make a comparison with other extraction techniques, but we acknowledge that this technique may underestimate the total Fe in mineral dust, as crystalline aluminium silicate minerals may not be fully digested. Consequently, the Fe solubility for the Libyan dust sample could be overestimated and even lower than that observed for the CFA samples. Page 5, lines 187-189 have been updated as follows:

"The recovery of Fe assessed using a standard reference material for urban particulate matter (NIST SRM 1648A) was around 89%. Therefore, the total Fe in the Libyan dust precursor sample could be underestimated somewhat as crystalline aluminium silicate minerals may not be fully digested".

**Comment 24: This is useful information but more explanation is needed here. You haven't mentioned ATD before. Explain what it is and why you used it. Could you also include the relative proportions of each Fe mineral phase in ATD in the SI? This data is of interest as more people start to use ATD as an SRM.**

Response: The sequential extraction techniques were tested using the Arizona Test Dust (ATD, Power Technology, Inc.). The ATD was also used as reference sample to check the recovery of the extracted Fe. A summary of the results for the ATD was added to the supporting information (Table S2).

Page 5, lines 190-192 have been updated as follows:

"The sequential extraction techniques were tested using the Arizona Test Dust (ATD, Power Technology, Inc.). The RSD% obtained for each extract using the ATD was 3% for FeA, 11% for FeD, 12% for FeM (n=7) and 2% for the 179 total Fe (n=3). A summary of the results for the ATD is reported in Table S2".

**Comment 25: The first 2-3 h are the time scale most critical to in-cloud processing. Could you address the rationale for the 168 h leaching in a little more detail? You could do a quick calculation to determine the solubility of Fe at pH 2.1 compared to 2.7 which allow you to make the argument that it is a pH effect rather than a solute concentration effect. If you do the suggested calculation you will be also be able to comment on the rate order of the observed dissolution kinetics.**

Response: The leaching experiments were conducted up to 168 h to better capture the dissolution curve in the kinetic model but also considering the tropospheric lifetime of aerosol particles. This is now stated in the manuscript at Page 7, lines 232-233.

At pH 2.7, the Fe solubility of Krakow ash measured after 2-hour leaching in 0.005 M $H_2SO_4$ - final pH 2.7 ($9 \cdot 10^{-5}$ mol $L^{-1}$) is lower than the equilibrium Fe solubility of highly reactive Fe species such as ferrihydrite ($6 \cdot 10^{-4}$ mol $L^{-1}$ from Shi et al. (2011a)), but higher than the solubility of more refractory Fe-bearing phases such as nanogoethite ($6 \cdot 10^{-6}$ mol $L^{-1}$ from Shi et al. (2011a)) and hematite ($1.6 \cdot 10^{-8}$ mol $L^{-1}$ from Bonneville et al. (2009)). This suggest that the system has reached the equilibrium as sufficient Fe may be dissolved from the highly reactive Fe species to suppress the dissolution of less reactive Fe. Page 7, lines 236-237 have been updated as follows:

"In the case of Krakow ash, the dissolution plateau was reached after 2-hour leaching in 0.005 M $H_2SO_4$ as sufficient Fe may be dissolved from the highly reactive Fe species to suppress the dissolution of less reactive Fe".

**Comments 26-27: Given in the main manuscript.**

Response: The reviewer comments have been implemented in the manuscript (Page 8, lines 279, 285)

**Comments 28-30: (28) Clarify if this the Libyan soil. (29) Slightly confusing labelling as Western Sahara is a country. If you can't change the label on the spectral plot, can you address this in the caption? (30) I'm not seeing the double peaks for the Icelandic and Saharan dusts in the pre-edge region. Why do you think there are more similarities with the Icelandic dust? Also sourced from high temp processes or a coincidence? Move Fig s2 into main text – you could have a Fig 5a and b.**

Response: The reviewer comments 28-30 have been implemented in the manuscript. Former Figures 5 and S2 have been merged as Figure 5. Figure 5b shows the pre-edge region at higher resolution. The Icelandic dust sample MIR-45 has been replaced with the sample D3 (Baldo et al., 2020). The mineral dust sample from western Sahara is now labelled "WS dust" on the spectral plot. We compared CFA and Icelandic dust because both have high content of aluminium silicate glass. Icelandic dust has volcanic origin as the parent sediments originate from the glacial erosion of volcanic deposits such as hyaloclastite (deposits consisting in glass fragments formed by rapid cooling of magna on contact with

water or ice), and lava flows (Baldo et al., 2020). On the other hand, CFA is formed due to rapid cooling of particles transported in flue gases during coal combustion (e.g., Jones, 1995).

Page 8, lines 292-306 have been updated as follows:

"The XANES Fe K-edge spectra of the CFA samples have some common features with those of Icelandic dust but tend to differ from mineral dust sourced in Saharan dust source region. In the pre-edge region of the spectrum, Icelandic dust (sample D3 in Fig. 5a-b) showed a main peak at around 7114.4 eV and a second less intense peak at around 7112.7 eV, while a broad peak was observed at around 7131.9 eV in the edge region (Baldo et al., 2020). A mineral dust sample from western Sahara (WS dust in Fig. 5a-b) showed a distinct double peak in the pre-edge region at around 7113.9 and 7115.2 eV, and a main peak in the edge region at around 7133.3 eV (Baldo et al., 2020). The similarities between Icelandic ash and CFA could be because aluminium silicate glass is dominant in these samples (e.g., Baldo et al., 2020; Brown et al., 2011), while Fe-bearing phases in mineral dust from the Saharan region are primarily iron oxides minerals such as hematite and goethite, clay minerals and feldspars (e.g., Shi et al., 2011b)".

**Comments 31: The model is struggling to capture the initial rapid dissolution relevant to in-cloud processing. Why do you think this is?**

Response: The laboratory measurements of the dissolution kinetics and Fe speciation were used to model the Fe solubility in the CFA samples. The dissolution of FeA as fast pool is insufficient to account for the initial rapid increase in Fe solubility, which indicate additional Fe was dissolved from more refractory Fe-bearing phases. Using FeM as intermediate pool improved the agreement between the model results and observations, but further work is needed to investigate the link between Fe solubility on Fe mineralogy. This is discussed in more details in section 5.1.

**Comments 32-34: Given in the main manuscript.**

Response: The reviewer comments have been implemented in the manuscript (Page 9, line 325 and Page 10, lines 361, 363)

**Comment 35: Are you saying that there is a question about the reliability of the field data from 29/12/08 or the whole transect? If the latter, this could be a problem. Have you looked at satellite AOD data for this date and the days either side to try to establish if this peak is likely to be reliable?**

Response: Unfortunately, it is hard to interpret the satellite images and AOD data available for this location during 27-30 December 2008. However, we have looked at total Fe concentrations on this date and the days before and after. The total Fe observed in $PM_{10}$ (430 ng m$^{-3}$) on 29 December 2008 is lower than that measured on the day before (667 ng m$^{-3}$) and the day after (773 ng m$^{-3}$), showing a

negative peak, as opposed to the positive peak in $PM_{2.5}$ (Srinivas et al., 2012). Thus, the extreme value recorded only for $PM_{2.5}$ on this date may be an outlier.

Page 11, lines 390-395 have been updated as follows:

"On the other hand, the model could not reproduce the peak in total Fe concentration (1.8% of Fe content in $PM_{2.5}$ sample) reported around 29 December 2008. The total Fe observed in $PM_{10}$ (430 ng m$^{-3}$) on 29 December 2008 is lower than that measured on the day before (667 ng m$^{-3}$) and the day after (773 ng m$^{-3}$), whereas that in $PM_{2.5}$ peaked on 29 December 2008 (Srinivas et al., 2012). Thus, the extreme value recorded only for $PM_{2.5}$ on this date may be an outlier".

**Comment 36: I wonder how familiar people are with Taylor diagrams. There is a lot of information presented in Fig. 9 but it's not very accessible. I suggest putting this plot in the SI.**

Response: The Taylor diagram in former Figure 9 (current Figure S3) has been moved to the supporting information and replaced with former Figure S7.

**Comments 37-38: Given in the main manuscript.**

Response: The reviewer comments have been implemented in the manuscript (Page 12, lines 421, 429)

**Comments 39-40: (39) Fu et al used a different leach technique. They didn't include any oxalate. This different leach media will almost certainly impact dissolution kinetics and fractional solubility. In addition, you have already noted that the different CFAs in your study have different kinetics. I suggest changing this to 'The Fe in our CFA samples initially dissolved faster than those used by Fu et al. (2012). This could be due to differences in the Fe speciation of the CFA samples in the two studies and/or the different leach media used.' – or words to that effect. (40) In Chen et al's Fig. 4, it was 6- 25% at pH 2 – this was the leach closest (but not the same) to your one. It was 21-70% after three pH cycles which is a very different leaching scheme to your one. The more appropriate comparison is with the data they show in Fig. 4 BUT you must note that your study and those of Fu and Chen all use quite different leach solutions and durations as some of the variability will result from these different conditions.**

Response: This is a very fair comment. What we did was to compare specifically the dissolution kinetics of CFA at pH 2 (acid only) observed in this study with data from the literature. We have now made this point clearer. Other conditions (e.g., with oxalate) were also discussed throughout section 5.1. We agree with the reviewer that the dissolution behaviour of CFA may be affected by the kind of CFA samples and/or the leaching media used in the different studies. We also wanted to highlight that at the same experimental conditions the CFA dissolution kinetics can vary considerably depending on the CFA type. Both of these points are made clearer in the revised text.

Page 12, lines 428-443 have been updated as follows:

"Our results showed a strong pH dependence in low ionic strength conditions, with higher dissolution rates at lower pH. For example, reducing the solution pH from 2.7 to 2.1, the Fe solubility of Krakow ash in $H_2SO_4$ only increased by a factor of 4 (Fig. 1a) over the duration of the experiments, while the Fe solubility of Aberthaw and Shandong ash increased by 9-10 times from pH 2.9 to pH 2.2 (Figs. 1b-c). This enhancement is higher than that observed in studies conducted on mineral dust samples, which showed that one pH unit can lead to 3-4 times difference in dissolution rates (Ito and Shi, 2016; Shi et al., 2011a). Furthermore, Chen et al. (2012) reported that the Fe solubility of the certified CFA 2689 only increased by 10% from pH 2 to pH 1, after 50 hours of dissolution in acidic media. The Fe solubility of CFA ($PM_{10}$ fractions) after 6 hours at pH 2 was 6%-10% for Aberthaw and Shandong ash respectively, and 28% for Krakow ash (Fig. 1). The Fe in our CFA samples initially dissolved faster than those used by Fu et al. (2012), who reported 2.9%-4.2% Fe solubility in bulk CFA from three coal-fired power plants in China after 12-hour leaching at pH 2. These results suggest that there are considerable variabilities in the pH dependent dissolution of Fe in CFA. This could be due to differences in the Fe speciation between CFA samples and/or the different leaching media used".

**Comment 41: Note that this is in contrast to Chen et al's findings**

Response: This was mentioned at line 456-457, Page 12:

"Cwiertny et al. (2008) reported that at pH 1-2 the high ionic strength generated by NaCl up to 1 M did not influence Fe dissolution of mineral dust particles".

**Comments 42-43: Given in the main manuscript.**

Response: The reviewer comments have been implemented in the manuscript (Page 12, line 449 and Page 14, line 503).

**Comments 44-45: (44) this is a contradiction – estimate with more accuracy perhaps? (45) 'More' More work to determine the mineralogy of Fe in 'natural' and processed CFAs, in combination with solubility experiments, in order to investigate the links between Fe solubility and Fe speciation/mineralogy is needed**

Response: Page 14, lines 523-527 have been updated as follows:

"In order to investigate the links between Fe solubility and Fe speciation/mineralogy, more work is needed to determine the Fe mineralogy in CFA samples at emission and after atmospheric processing, in combination with solubility experiments".

**Comment 46: I really think that the plots of your data that are currently in the SI should be moved into the main manuscript, as you refer to them quite extensively and it is much easier to follow if you don't have to keep flicking between the SI and manuscript. These plots could be consolidated, e.g., S8 and S9 could become one plot. Similarly, S3-5 could also become one plot. The tables should stay in the SI.**

Response: As suggested former Figures S3, S4 and S5 have been merged as Figure 7 and former Figures S8 and S9 have been combined in Figure 10.

**Comments 47-48: (47) Libyan rather than Libya. Change Saharan to Libyan to save confusion. (48) I would assume that most people don't know where Tibesti is. I certainly didn't! State that the sample is also from a Saharan dust source region**.

Response: Page 15, lines 555-564 have been updated as follows:

"High ionic strength also impacted the dissolution rates of the Libyan dust precursor sample at low pH (Fig. S4). At around pH 2 conditions, the proton-promoted Fe dissolution of Libyan dust was enhanced by ~40% after the addition of $(NH_4)_2SO_4$. At around pH 2 and with 0.01 M $H_2C_2O_4$, the Fe solubility of Libyan dust decreased by ~30% in the presence of $(NH_4)_2SO_4$. Overall, the Fe solubility of Libyan dust was lower compared to that observed in the CFA samples. After 168 hour-leaching at pH 2.1 with 1 M $(NH_4)_2SO_4$, the Fe solubility of Libyan dust was 7.2% (Fig. S4), which was from around 3 to 7 times lower compared to that of the CFA samples (Fig. 1). At around pH 2 conditions in the presence of oxalate and high $(NH_4)_2SO_4$ concentration, the Fe solubility of Libyan dust rose to ~13.6% (Fig. S4), which is still 4 times lower than that of Krakow ash and around 1.5 lower than Aberthaw and Shandong ash (Fig. 2). The Fe solubilities of the Libyan dust observed in this study are comparable with those of the Tibesti dust (Tibesti Mountains, Libya, 25.583333N/16.516667E) in Ito and Shi (2016) at similar experimental conditions".

**Comment 49: Can you quantify this? Yes, you have more of a range of concentrations but many (most?) of the model output data looks to be different than the observation.**

Response: Former Figure 10 has been moved to the supporting information (current Figure S5) and updated with the aerosol Fe solubility data.

Page 15, lines 583-586 have been updated as follows:

"In Fig. S5, the model estimates of aerosol Fe solubility over the Bay of Bengal considerably improved in Test 1 (RMSE 11) compared to Test 0 (RMSE 21), but more work is needed to improve size-resolved Fe emission, transport, and deposition".

**Comments 50-54: Given in the main manuscript.**

Response: The reviewer comments have been implemented in the manuscript (Page, 16 lines 587, 591, Page 18 lines 627, 628, Page 19 line 630).

**Comments 55-56: (55) Add amorphous Fe to FeA and goethite/hematite Fe to FeD, and other Fe (including Fe in alumino-silicates). (56)** Given in the main manuscript.

Response: Former Figures 5 and S2 have been merged as Figure 5 in the revised manuscript. Figure 5b shows the pre-edge region at higher resolution. The Icelandic dust sample MIR-45 has been replaced with the sample D3 (Baldo et al., 2020).

The caption of Figure 5 has been updated as follows:

"Figure 1: Fe speciation in CFA and mineral dust samples. a-b) Fe K-edge XANES spectra of Krakow ash, Aberthaw ash, magnetite, hematite, and illite standards, mineral dust from the Dyngjusandur dust hotspot in Iceland - D3 (Baldo et al., 2020), and mineral dust from western Sahara - WS dust (Shi et al., 2011b). c) Percentages of ascorbate Fe (amorphous Fe, FeA), dithionite Fe (goethite/hematite, FeD), magnetite Fe (FeM), and other Fe (including Fe in aluminosilicates) to the total Fe (FeT) in the CFA samples and Libyan dust precursor. The FeT (as %wt.) is given below each sample column. The data uncertainty was estimated using the error propagation formula: 4% for FeA/FeT, 11% for FeD/FeT, 12% for FeM/FeT, and 2 % for FeT."

**Comments 57: Part of this reference is missing.**

Response: This has been corrected:

Pye, H. O. T., Nenes, A., Alexander, B., Ault, A. P., Barth, M. C., Clegg, S. L., Collett Jr, J. L., Fahey, K. M., Hennigan, C. J., Herrmann, H., Kanakidou, M., Kelly, J. T., Ku, I. T., McNeill, V. F., Riemer, N., Schaefer, T., Shi, G., Tilgner, A., Walker, J. T., Wang, T., Weber, R., Xing, J., Zaveri, R. A., and Zuend, A.: The acidity of atmospheric particles and clouds, Atmos. Chem. Phys., 20, 4809-4888, doi: 10.5194/acp-20-4809-2020, 2020.

**Referee 2 – Morgane Perron**

General Response: We thank Dr Perron for the valuable and comprehensive comments. The comments have been addressed point by point below.

**Comment 1: This hasn't been shown (to this extent) by any study. In order to have such impact CFA emissions would have to be both highly soluble (they are) and of very large magnitude (not shown to date). Maybe just pace down the sentence by choosing another word.**

Response: Page 1, lines 14-16 have been updated as follows:

"Mineral dust is the largest source of aerosol iron (Fe) to the offshore global ocean, but acidic processing of coal fly ash (CFA) in the atmosphere could be an important source of soluble aerosol Fe".

**Comment 2: Associated figure says "other" I cannot recall any data showing the remaining fraction is aluminosilicate. If no proof maybe correct to "remaining 50%-87 % of Fe was thought to be associated with ..." or "remaining 50%-87 % of Fe may be comprised in …"**

Response: Page 1, lines 19-20 has been updated as follows:

"The remaining 50%-87 % of Fe was associated with other Fe-bearing phases, possibly aluminosilicates".

**Comment 3-4: (3) Either because of the pH or because of differences in the samples…? I'd delete this sentence this is not the main finding (4) No word on oxalate addition?**

Response: Page 1, lines 21-24 have been updated as follows:

"The oxalate effect on the Fe dissolution rates at pH 2 varied considerably depending on the samples, from no impact for Shandong ash to doubled dissolution for Krakow ash".

**Comment 5: Given in the main manuscript.**

Response: Page 1, lines 27-28 have been updated as follows:

"Overall, Fe in CFA dissolved up to 7 times faster than in a Saharan dust precursor sample at pH 2".

**Comment 6: I do not think the model focuses on surface Fe dissolution only. Likely it accounts for Fe dissolution form one aerosol as a whole (although it is thought that surface Fe will dissolve first).**

Response: We did not mean the particulate surface by surface, but the mass concentration in the air over the surface ocean. We corrected it.

Page 1, lines 30-34 have been updated as follows:

"The revised model showed a better agreement with observations of Fe solubility in aerosol particles over the Bay of Bengal, due to the initial rapid release of Fe and the suppression of the oxalate-promoted dissolution at low pH"

Page 10, lines 373-379 have been updated as follows:

"Observations of total Fe concentration and Fe solubility in $PM_{2.5}$ along the cruise tracks over the Bay of Bengal for the period extending from 27 December 2008 to 26 January 2009 (Bikkina et al., 2020) were compared with temporally and regionally averaged data from model estimates. The daily averages of model results were calculated from hourly mass concentrations in the air over the surface ocean along the cruise tracks."

**Comment 7: Community now tries to use "bioaccessible" Fe as the genuine bioavailability is still to prove.**

Response: The term "Bioavailable Fe" has been replaced with "Bio-accessible Fe" throughout the manuscript.

**Comment 8: Given in the main manuscript.**

Response: The reviewer comment has been implemented in the manuscript (Page 2, line 50).

**Comments 9-10: (9) I would say <1% is more accurate according to published literature. I'd suggest citing Sholkovitz et al 2012 and Sholkovitz et al 2009. (10) Reference needed.**

Response: The reviewer comments have been implemented in the manuscript and the references updated (Page 2, lines 53 and 58).

**Comment 11: Given in the main manuscript.**

Response: The Fe solubility from laboratory measurements on pyrogenic sources is reported in Table 2 of Ito et al. (2021b).

**Comments 12-14: Given in the main manuscript.**

Response: The reviewer comments have been implemented in the manuscript.

Page 2, lines 61-64 have been updated as follows:

"Wang et al. (2015) estimated that coal combustion emitted around ~0.9 Tg yr$^{-1}$ of Fe into the atmosphere (on average for 1960–2007), contributing up to ~86% of the total anthropogenic Fe emissions. A more recent study, which has included metal smelting as an atmospheric Fe source, estimated that coal combustion emitted ~0.7 Tg yr$^{-1}$ of Fe for the year 2010, contributing around 34% of the total anthropogenic Fe atmospheric loading (Rathod et al., 2020)".

**Comment 15: Given in the main manuscript.**

Response: The reviewer comment has been implemented in the manuscript (Page 2, line 70).

**Comment 16: Given in the main manuscript.**

Response: Page 3, lines 78-79 have been updated as follows:

"As a result, a thin layer of water with high acidity, low pH and high ionic strength is formed around the particles (Meskhidze et al., 2003; Spokes and Jickells, 1995; Zhu et al., 1992)".

**Comment 17: Given in the main manuscript.**

Response: We are referring to aged aerosols including aged CFA particles. The reviewer comment has been implemented in the manuscript (Page 3, line 82).

**Comment 18: Oxalic acid? Maybe spell out the name when first encountered?**

Response: Please, note that the name and formula of oxalic acid are given at line 77, Page 3.

**Comment 19: Given in the main manuscript.**

Response: The reviewer comment has been implemented in the manuscript (Page 3, lines 88-89).

**Comment 20: Given in the main manuscript.**

Response: The reviewer comment has been implemented in the manuscript (Page 3, line 96).

**Comment 21: Given in the main manuscript.**

Response: The reviewer comment has been implemented in the manuscript (Page 3, line 101).

**Comment 22: called 'CFA' up to then.**

Response: The reviewer comment has been implemented throughout the manuscript.

**Comment 23: Is the term 'phase' commonly used to refer to the mineralogy of the particles holding Fe?**

Response: the term Fe phase has been replaced with Fe-bearing phase throughout the manuscript.

**Comment 24: I would pace that down. 'confirming the existence of different Fe-bearing mineral 'phases' within a single CFA sample.**

Response: Page 4, lines 115-118 have been updated as follows:

"Previous studies showed that CFA dissolves much faster during the first 1-2 hours than subsequently (Borgatta et al., 2016; Chen et al., 2012; Chen and Grassian, 2013; Fu et al., 2012; Kim et al., 2020), confirming the existence of multiple Fe-bearing phases within a single CFA sample with different dissolution behaviour".

**Comments 25-26: Given in the main manuscript.**

Response: Page 4, lines 119-120 have been updated as follows:

"In this study, laboratory experiments were conducted to determine the dissolution kinetics of coal combustion emission products (i.e., CFA)"

**Comments 27-28: (27) I am wondering about the use of the word "dust" sometimes in the manuscript. "Dust" would refer to a lithogenic mineral source (wind-blown soil). Maybe dust here would rather be changed to "aerosol fractions" which is the generic term for 'particulates suspended in the air' (28) Change to "aerosols".**

Response: The reviewer comment has been implemented throughout the manuscript. The term dust is now used specifically for lithogenic sources.

**Comments 29 (see also comment 23 Referee 1): No HF used for digestion? Did you check the recovery, especially for the dust material (Libya dust and ATD)?**

Response: For the analysis of the total Fe content, we only used microwave digestion in concentrated $HNO_3$. The recovery of Fe was around 89% which was assessed using a standard reference material for urban particulate matter (NIST SRM 1648A). We did not make a comparison with other extraction techniques, but we acknowledge that this technique may underestimate the total Fe in mineral dust, as crystalline aluminium silicate minerals may not be fully digested. Consequently, the Fe solubility for the Libyan dust sample could be overestimated and even lower than that observed for the CFA samples. Page 5, lines 187-189 have been updated as follows:

"The recovery of Fe assessed using a standard reference material for urban particulate matter (NIST SRM 1648A) was around 89%. Therefore, the total Fe in the Libyan dust precursor sample could be underestimated somewhat as crystalline aluminium silicate minerals may not be fully digested".

**Comment 30 (see also comment 24 Referee 1): Given in the main manuscript.**

Response: The sequential extraction techniques were tested using the Arizona Test Dust (ATD, Power Technology, Inc.). The ATD was also used as reference sample to check the recovery of the extracted Fe. A summary of the results for the ATD was added to the supporting information (Table S2).

Data are reported as average and standard deviation of n replicates.

Page 5, lines 190-192 have been updated as follows:

"The sequential extraction techniques were tested using the Arizona Test Dust (ATD, Power Technology, Inc.). The RSD% obtained for each extract using the ATD was 3% for FeA, 11% for FeD, 12% for FeM (n=7) and 2% for the 179 total Fe (n=3). A summary of the results for the ATD is reported in Table S2".

**Comment 31: May I suggest a summary table in the manuscript reporting each pH, H2SO4 concentration, addition of H2C2O4 and NH4SO4 would make the reading much easier.**

Response: Table S1 reports the concentrations of $H_2SO_4$, $H_2C_2O_4$ and $(NH_4)_2SO_4$ in the experiment solutions, the original and final pH from model estimates (including $H^+$ concentrations and activities), and the pH measurements for the solution with low ionic strength. The concentrations of $H_2SO_4$, $H_2C_2O_4$ and $(NH_4)_2SO_4$ in the experiment solutions and the final pH from model estimates are also reported in the legends of all figures.

**Comment 32: On Fig 4 it looks like the addition of (NH4)2SO4 does slightly (yet noticeably) diminish the Fe dissolution. Or is that another factor coming into play?**

Response: This was discussed in section 5.1, Page 13, lines 494-499:

"We speculate that the high concentration of sulfate ions is likely to be responsible for inhibiting the oxalate-promoted dissolution by reducing oxalate adsorption on the particle surface. At pH 1 in the presence of oxalate, increasing the concentration of $(NH_4)_2SO_4$ from 0.5 M to 1.5 M did not affect the Fe dissolution behaviour of the CFA samples (Fig. 4). As previously discussed, the adsorption of sulfate ions on the particle surface may inhibit oxalate-promoted dissolution. However, once the saturation coverage is reached, increasing the concentration of anions has no further effect on the dissolution rate (Cornell et al., 1976)".

**Comment 33: Given in the main manuscript.**

Response: Libyan dust precursor is now specified throughout the manuscript.

**Comments 34-38 (see also comments 28-30 Referee 1): (34) Please develop slightly the reasoning here. (35) This is a funny parallel between CFA and dust, why would similarities be expected? Is that in order to better understand the composition (mineralogy) of the CFA? Is the exact composition of the Icelandic dust known? Is it dust or aerosols collected at a random location in**

**Iceland? Maybe a word to understand why the parallel between CFA and dust is made would help. (36) Libya dust? (37) Libya dust? Please keep wording consistent (38) I cannot see the double peak in the pre-edge region though I do not have a trained eye on this kind of graph. Maybe it isn't that distinct?**

Response: The chemical and mineralogical composition of Icelandic dust $PM_{10}$ fractions from 5 major dust hotspots in Iceland is reported in Baldo et al. (2020). We decided to make a comparison between Icelandic dust and CFA because the dominant component in Icelandic dust is aluminium silicate glass which is also a primary component of CFA (e.g., Brown et al., 2011). Icelandic dust has volcanic origin as the parent sediments originate from the glacial erosion of volcanic deposits such as hyaloclastite (deposits consisting in glass fragments formed by rapid cooling of magna on contact with water or ice), and lava flows (Baldo et al., 2020). On the other hand, CFA is formed due to rapid cooling of particles transported in flue gases during coal combustion (e.g., Jones, 1995),

Former Figures 5 and S2 have been now merged as Figure 5. Figure 5b shows the pre-edge region at higher resolution, increasing the resolution the double peak should be more visible. The Icelandic dust sample MIR-45 has been replaced with the sample D3 (Baldo et al., 2020).

Page 8, lines 292-306 have been updated as follows:

"The XANES Fe K-edge spectra of the CFA samples have some common features with those of Icelandic dust but tend to differ from mineral dust sourced in the Saharan dust source region. In the pre-edge region of the spectrum, Icelandic dust (sample D3 in Fig. 5a-b) showed a main peak at around 7114.4 eV and a second less intense peak at around 7112.7 eV, while a broad peak was observed at around 7131.9 eV in the edge region (Baldo et al., 2020). A mineral dust sample from western Sahara (WS in Fig. 5a-b) showed a distinct double peak in the pre-edge region at around 7113.9 and 7115.2 eV, and a main peak in the edge region at around 7133.3 eV (Baldo et al., 2020). The similarities between Icelandic ash and CFA could be because aluminium silicate glass is dominant in these samples (e.g., Baldo et al., 2020; Brown et al., 2011), while Fe-bearing phases in mineral dust from the Saharan region are primarily iron oxides minerals such as hematite and goethite, clay minerals and feldspars (e.g., Shi et al., 2011b)".

**Comment 39 (see also comment 31 Referee 1): In Fig 6a,c,d , the model seems to miss out on the rapid increase in Fe solubility (as if the solubility of Fe in the model had a threshold up to 24h-time, the latter which seem bias the model outcome). On the contrary the subsequent Fe dissolution (after 24h) seem exaggerated in the model compared to the slow increase in the experimental result. A word on this?**

Response: The laboratory measurements of the dissolution kinetics and Fe speciation were used to model the Fe solubility in the CFA samples. The dissolution of FeA as fast pool is insufficient to account for the initial rapid increase in Fe solubility, which indicate additional Fe was dissolved from more refractory Fe-bearing phases. Using FeM as intermediate pool improved the agreement between the

model results and observations, but further work is needed to investigate the link between Fe solubility on Fe mineralogy. This was discussed in more detail in section 5.1, Page 14, lines 500-527.

**Comment 40: Given in the main manuscript.**

Response: The reviewer comment has been implemented in the manuscript (Page 9, line 325).

**Comment 41: Would a table help better understanding the different model test-runs?**

Response: Table S4 with the modelled mass concentrations of total Fe and Table S5 with the modelled aerosol Fe solubility have been added to the supporting information. Please note that the Fe speciation, the measurements of the Fe dissolution kinetics, and the results of the IMPACT model for each sensitivity simulation (Test 0-3) can be downloaded at:

https://doi.org/10.25500/edata.bham.00000702.

**Comments 42-43: Given in the main manuscript.**

Response: The reviewer comments have been implemented in the manuscript (Page 10 lines 363 and 370).

**Comments 44-46: (44) do you mean total surface aerosol Fe? Vs surface dissolved Fe? The term aerosol Fe is confusing as dissolved Fe also refers to Fe which is part of aerosol particles. (45) How did measurements differentiate surface aerosol Fe to total aerosol Fe and surface dissolved aerosol Fe to total dissolvable aerosol Fe? Are all the measurements really specific to the surface of aerosols or is dissolved Fe only assumed to originate from the surface of aerosol (what is the aerosol Fe component then?)? Maybe better wording for each measurement is required for enhanced clarity. (46) Total Fe in aerosols I assume?**

Response: The terms "aerosol Fe" and "aerosol dissolved Fe" have been replaced throughout the manuscript with "mass concentration of total Fe" and "mass concentration of dissolved Fe", respectively.

Page 10, lines 373-385 have been updated as follows:

"Observations of total Fe concentration and Fe solubility in $PM_{2.5}$ along the cruise tracks over the Bay of Bengal for the period extending from 27 December 2008 to 26 January 2009 (Bikkina et al., 2020) were compared with temporally and regionally averaged data from model estimates. The daily average of model results were calculated from hourly mass concentrations in the air over the surface ocean along the cruise tracks The concentration of total Fe observed over the Bay of Bengal varies from $145 \pm 144$ ng m$^{-3}$ over the North Bay of Bengal (27 December 2008 - 10 January 2009) to $55 \pm 23$ ng m$^{-3}$ over the South Bay of Bengal (11-26 January 2009) (Bikkina et al., 2020). In Fig. 8, the modelled concentrations of total Fe exhibit a similar variability to that of measurements with relatively higher values over the North Bay of Bengal ($59 \pm 29$ ng m$^{-3}$ in different sensitivity simulations) compared to

the South Bay of Bengal ($20 \pm 12$ ng m$^{-3}$ in different sensitivity simulations). However, the modelled concentrations of total Fe were underestimated by a factor of $2.9 \pm 1.5$."

Please also note that in section 4.2 the description of the model outputs has been updated with the new model results.

**Comment 47: May I suggest that biomass burning is represented in orange or another colour as we cannot see it well as it currently is. Also, maybe instead of / in addition to plotting date as x axis, the location could be stated/ indicated (from the Northern Bay of Bengal to the South).**

Response: Former Figure 7 (current Figure 8) has been modified following the suggestion of the reviewer. In the caption, we added the reference time intervals for the measurements over the North and South Bay of Bengal.

**Comment 48 (see also comment 35 Referee 1): This peak is quite large for reflecting measurement uncertainty, isn't it? Have you checked whether this point may be the result of a specific weather event (fire emission for example, such a peak may be visible from satellite observation) or may originate from a single sample contamination (in this case it should be discarded)**

Response: Unfortunately, it is hard to interpret the satellite images and AOD data available for this location during 27-30 December 2008. However, we have looked at total Fe concentrations on this date and the days before and after. The total Fe observed in PM$_{10}$ (430 ng m$^{-3}$) on 29 December 2008 is lower than that measured on the day before (667 ng m$^{-3}$) and the day after (773 ng m$^{-3}$), showing a negative peak, as opposed to the positive peak in PM$_{2.5}$ (Srinivas et al., 2012). Thus, the extreme value recorded only for PM$_{2.5}$ on this date may be an outlier. Since we focus on Fe solubility in the revised paper, discarding this data point does not change the results significantly (see Figure R1).

Page 11, lines 390-395 have been updated as follows:

"On the other hand, the model could not reproduce the peak in total Fe concentration (1.8% of Fe content in PM$_{2.5}$ sample) reported around 29 December 2008. The total Fe observed in PM$_{10}$ (430 ng m$^{-3}$) on 29 December 2008 is lower than that measured on the day before (667 ng m$^{-3}$) and the day after (773 ng m$^{-3}$), whereas that in PM$_{2.5}$ peaked on 29 December 2008 (Srinivas et al., 2012). Thus, the extreme value recorded only for PM$_{2.5}$ on this date may be an outlier".

[Figure]

**Figure R1: Aerosol Fe solubility over the North Bay of Bengal obtained discarding data on 29 December 2008.**

**Comment 49: dFe concentrations in aerosols seem to fluctuate a lot throughout the sampling transect. I would personally use median value rather than average for aerosols to avoid being biased by very low/high data points. The lowest dissolved Fe data are towards the right-hand side of the graph which I understood as "southern Bay of Bengal" but with no spatial indication on the graph I may have mis**

Response: Please note that in section 4.2 the description of the model outputs has been updated with the new model results. This paragraph has been deleted as we now focus on the aerosol Fe solubility.

**Comment 50: I would personally have chosen Fig S7 to display on the ms instead of Fig 9. Indeed, Fig S7 as it shows the performance of each test-run to reproduce the observational data. Fig 9 is great but a little complex to interpret and to assess the model performance from.**

Response: Former Figure S7 has been moved into the manuscript (current Figure 9) while former Figure 9 is now in the supporting information (current Figure S3).

**Comments 51-53 (see also comments 39-40 Referee 1): (51) Different samples and leaching protocols and leaching solutions will also play a role in the differences highlighted here and above. 3 CFAs in this study already have different reactions to the same protocol, likely due to different mineralogy and different degree of combustion (this study has not addressed possible incomplete combustion from the sampled power plants). (52) Please see comment above, I believe many factors come into play and this conclusion cannot be drawn without acknowledging these**

parameters. **(53) Here may also be a good time to mention other parameters that come into play apart from pH. Reminding the readers the all these studies are not strictly comparable to one another.**

Response: The reviewer comment has been implemented in the manuscript.

Page 12, line 443 has been updated as follows:

"This could be due to differences in the Fe speciation between CFA samples and/or the different leaching media used."

**Comment 54: Is that the percentage increase or the final Fe solubility. Maybe the "from x%…to x%" is less prone to misunderstanding.**

Response: In this case, we are referring to the percentage increase in Fe solubility. The range of Fe solubilities is described in detail in the experiment results, section 3.1.

**Comment 55: Given in the main manuscript.**

Response: Page 12, lines 453-454 have been updated as follows:

"This may increase the surface negative charge favouring the absorption of $H^+$ and thereby increase Fe dissolution at the particle surface".

**Comment 56-57: Given in the main manuscript.**

Response: The reviewer comment has been implemented in the manuscript (Page 13, lines 472-473 and 486).

**Comment 58: Given in the main manuscript**

Response: Page 13, lines 491-492 have been updated as follows:

"The adsorption of anions can reduce oxalate adsorption on the particle surface due to electrostatic repulsion which results in slower release of Fe (Eick et al., 1999)".

**Comment 59: Less soluble phases? I believe they are all chemically stable but show different reactivity toward the leaching media**

Response: Page 14, lines 507-508 have been updated as follows:

"As the dissolution continued, more refractory phases became the dominant source of dissolved Fe (Shi et al., 2011a).".

**Comment 60 (see also Comments 44-45 of Referee 1): What is a pristine CFA? Aren't CFAs always the result of a combustion process (therefore, 'processed' particles)? Not sure about the wording used.**

Response: The term pristine CFA refers to the particles before undergoing atmospheric processing.

Page 14, lines 523-527 have been updated as follows:

"In order to investigate the links between Fe solubility and Fe speciation/mineralogy, more work is needed to determine the Fe mineralogy in CFA samples at the emission and after atmospheric processing, in combination with solubility experiments."

**Comment 61: Given in the main manuscript.**

Response: Former Figures S8 and S9 have been merged as Figure 10.

**Comment 62: Given in the main manuscript.**

Response: In section 5.2, we discussed the results for the experiment conducted on the Libyan dust precursor, which is now specified in the text.

**Comment 63: Could you provide the solubility in "" please so the reader has an idea of the impact of each treatment?**

Response: The Fe solubility % for each treatment are provided in the following paragraph, lines 559-562, Page 15. Please note that the Fe speciation, the measurements of the Fe dissolution kinetics can be also downloaded at: https://doi.org/10.25500/edata.bham.00000702.

**Comment 64: Given in the main manuscript.**

Response: The reference "Bikkina et al. (2020)" was added (Page 15, line 576).

**Comment 65: This source wasn't assessed in this study. Moreover, the sentence just above raises awareness on the need for more investigation on BB. The revised model has improved modelling of anthropogenic aeolian Fe dissolution.**

Response: This is true for the samples used in the dissolution experiments. However, the improvement of model-observation agreement supports the suppression of the oxalate-promoted dissolution found in this study. This was treated in Test 0, by assuming that the dissolution rate is independent from the pH for extremely acidic solutions (pH <2).

Page 15, lines 579-581 have been updated as follows:

"The revised model also enabled us to predict sensitivity to a more dynamic range of pH changes, particularly between anthropogenic combustion and biomass burning, by the suppression of the oxalate-promoted dissolution at pH lower than 2. In Test 0, the dissolution rate was assumed to be independent from the pH for extremely acidic solutions (pH <2)".

**Comment 66: Did improve yes but the Test 1 model run is still quite off for a range of datapoints**

Response: This is true, more work is needed to improve Fe emissions. Former Figure 10 has been moved to the supporting information (current Figure S5) and updated with the aerosol Fe solubility data .

Page 15, lines 583-586 have been updated as follows:

"In Fig. S5, the model estimates of aerosol Fe solubility over the Bay of Bengal considerably improved in Test 1 (RMSE 11) compared to Test 0 (RMSE 21), but more work is needed to improve size-resolved Fe emission, transport, and deposition".

**Comments 67 and 71: (67) Anthropogenic? (71) Is that only the anthropogenic fraction of pyrogenic Fe? Maybe stick to CFA-Fe or anthropogenic-Fe if more accurate? I would suggest to change the caption to 'Contribution of pyrogenic aerosols to the atmospheric dissolved Fe loading…'**

Response: We separated the contribution of anthropogenic combustion sources (ANTHRO) and biomass burning (BB) in Figure 11 and Figure 12, respectively.

Page 15, lines 586-588 have been updated as follows:

"The model results in Test 1 indicate a larger contribution of anthropogenic combustion sources to the atmospheric Fe loading over East Asia (Fig. 11), but a smaller contribution of biomass burning sources downwind from tropical regions (Fig. 12)."

The figure captions have been also updated as follows:

"Percentage contribution of anthropogenic combustion/biomass burning aerosols to the atmospheric dissolved Fe loading…"

**Comment 68: No short conclusion?**

Response: As the main findings were already summarised in the discussion part, we think that adding a conclusion would be a repetition of the discussion and of the abstract.

**Comment 69: BB could have been another colour (orange for example) for a better visual.**

Response: Former Figure 8 showing the dissolved Fe concentrations over the Bay of Bengal has been deleted as we now focus on the aerosol Fe solubility.

**Comment 70: Can the colour scale be improved in the dark blue and purple shades?**

Response: The reviewer comment has been implemented in current Figure S5.

**Comment 72: Maybe Figures 10 and 11 could go to SI as they are only very briefly discussed**

Response: Former Figure 10 has been update un moved to the suppurting information (current Figure S5).

**Figures and Tables**

Former Figures 7-11 and Figures S2-S10 have been rearranged following the suggestion of the reviewers. Former Figure 8 was deleted as we focus on the aerosol Fe solubility.

Additional Tables were provided in the supporting information according to the comments of the reviewers.

**List of new Tables: Tables S2, S4, S5**

**Table S1: Percentages of ascorbate Fe (FeA), dithionite Fe (FeD), magnetite Fe (FeM), and total Fe (FeT) in the Arizona Test Dust (ATD, Power Technology, Inc.).  to the total dust mass (wt%). For each type of extracted Fe, the standard deviation (sd) and number of replicates (n) is reported.**

| Fe species | wt% | sd | n |
|---|---|---|---|
| FeA | 0.057 | 0.002 | 7 |
| FeD | 0.394 | 0.045 | 7 |
| FeM | 0.047 | 0.006 | 7 |
| FeT | 3.501 | 0.056 | 3 |

**Table S2: Modelled mass concentration of total Fe in PM$_{2.5}$ aerosol particles (ng m$^{-3}$) over the Bay of Bengal from 27 December 2008 to 26 January 2009. Observations are reported in Bikkina et al. (2020). The concentrations of total Fe were calculated along the cruise tracks in the North Bay of Bengal (27 December 2008 - 10 January 2009) and the South Bay of Bengal (11-26 January 2009) using the IMPACT model. The total Fe emissions from anthropogenic combustion sources (ANTHRO) and biomass burning (BB) were estimated using the emission inventory of (Ito et al., 2018), whereas Fe emissions from mineral dust sources (DUST) were dynamically simulated (Ito et al., 2021a).**

[revised manuscript text omitted]

---

## Author Response (AR2)

**Response to the additional comments from referee 1 – Dr Rachel Shelley**

General Response: We thank Dr Shelley for supporting the findings of this manuscript. The comments have been addressed point by point below.

**Comment at Line 48: As you give the solubility as a percentage, the equation should be dissolved Fe/ total Fe x 100**

Response: The reviewer comment has been implemented in the manuscript

**Comment at Line 55. Put a comma between sources and with**

Response: This has been revised.

**Comment at Lines 179-182. 89% recovery is a bit low. Usually, you would want at least 90% to demonstrate good recovery. Morton et al. (2013) demonstrated that HF was required for full recovery of marine aerosols. Rather than saying around 89% it would be better to give a mean and standard deviation. In addition, to the CRM and the Saharan soil, it is possible that the CFAs wouldn't have fully digested, although I am slightly reassured by your good recovery for the ATD. It would be worth highlighting this. If you contact Peter Morton (pmorton@fsu.edu), he will be able to provide you with the latest consensus value for total Fe in the ATD. For this reason (ATD recovery), and because you have now stated that fractional solubility might be slightly over-estimated and demonstrated that your Fe dissolution scheme has improved the performance of the IMPACT model, this study should be published. However, before publication I'd like to see more detail about the digestions, such as the model and the programme used (times, temp and pressure – could be a table in the supplement) and the masses of the material digested included in the manuscript.**

Response: As suggested, a detailed description of the digestion method used in this study was added to the supporting information (Text S1). Dr Peter Morton has provided us with the latest up-to-date average Fe content measured in ATD ($33,039 \pm 3,834$ ug/g). The estimated recovery of Fe from the ATD samples calculated using the reference total Fe in ATD (from Dr Morton) and the total Fe measured in this study using the microwave digestion method (for the calculation we used the total Fe values prior the correction for the Fe recovery based on the NIST results) is $94.0\% \pm 1.5\%$. Although Fe recovery from NIST was slightly lower than 90% ($89.0\% \pm 0.4\%$), the ATD samples showed good recovery. This has been highlighted in the text. We agree that ideally the recovery would be better to be over 90%. We feel that the reliability of the recovery is also more important. 89% recovery is indeed very good in analytical chemistry. We also noted that the uncertainty associated with this recovery is more than an order of magnitude lower than model simulations of fractional Fe solubility. We therefore suggest that such uncertainty does not affect the conclusion of the study.

Lines 179-187 have been updated as follows:

"The total Fe content in the samples was determined by microwave digestion in concentrated nitric acid ($HNO_3$) followed by inductively coupled plasma mass spectrometry (ICP-MS) analysis. A detailed description of the digestion method is provided in the supporting information (Text S1). The total Fe content obtained for the Arizona Test Dust (ATD, Iso 12103-1, Power Technology, Inc.) was comparable with the latest consensus value for the total Fe in ATD which indicates a good recovery (94.0% ± 1.5%). The recovery of Fe assessed using a standard reference material for urban particulate matter (NIST SRM 1648A) was 89.0% ± 0.4%. It is possible that some of the Fe in aluminosilicate minerals are not fully digested but the uncertainty associated with this analytical method is very small, particularly when we compare this with the large uncertainty in simulated Fe solubility in models".

Description of the microwave digestion method added to the supporting information as Text S1:

"1-3 mg of dust/ash were weighed on quartz filters (1.5 $cm^2$ punch). The filters were placed into vessels with 10 ml of 68% ultrapure nitric acid ($HNO_3$, Romil). The vessels were loaded into a MARS 6 Microwave Digestion System (CEM Technology). The filter-membrane programme was used to digest the samples. This consists of an increase in temperature to 200°C (15-min ramp time) followed by 15 min at 200°C and pressure 800 psi. The sample solutions were then diluted to 2% $HNO_3$ and filtered through 0.45 µm membrane filters. The samples were stored in the fridge at 4°C prior the analysis. The Fe concentration in the filtrates was measured by inductively coupled plasma mass spectrometry (ICP-MS) analysis.

After each digestion, the vessels were cleaned to prevent contamination. These were first washed with DI water and dried in the oven at 70°C. Subsequently, 10 ml of concentrated $HNO_3$ was added to each vessel which were placed into the microwave to undergo the cleaning programme (ramp time of 15 min to 190°C, followed by 10 min at 190°C and pressure 800 psi). Finally, the vessels were rinsed with DI water and air-dried in a fume hood. All glassware was acid washed in 10% $HNO_3$.

To assess the recovery of Fe, we used a standard reference material for urban particulate matter (1 mg of NIST SRM 1648A on quartz filter). The recovery of Fe from NIST was 89.0% ± 0.4%. We used the Arizona Test Dust (ATD, Iso 12103-1, Power Technology, Inc.) to test the method. The estimated total Fe content in the ATD was 3.501% ± 0.056% which is comparable with the latest consensus value for the total Fe in ATD. Here, the estimated recovery of Fe from the ATD samples calculated using the reference total Fe in ATD and the total Fe in the ATD samples obtained in this study (prior the correction for the Fe recovery based on the NIST results) is 94.0% ± 1.5%".

**Comment at Lines 183-185. Could you add the wt%s of Fe in ATD and the standard deviations that are in table S2 to the text and move the RSDs to the table as the abundances of each fraction are of more interest than the RSDs. You should also include your percent recovery, as ATD is a community-consensus reference material.**

Response: The reviewer comment has been implemented in the manuscript.

The Fe recovery from ATD was added to the text and supporting information (Text S1) and Lines 188-191 have been updated as follows:

"The sequential extraction techniques were tested using the ATD. The wt% of Fe obtained for each extract using the ATD was $0.057 \pm 0.002$ for FeA, $0.394 \pm 0.045$ for FeD, $0.047 \pm 0.006$ for FeM (n=7) and $3.501 \pm 0.056$ for the total Fe (n=3). A summary of the results for the ATD is reported in Table S2."

**Comment at Line 287. …the western Sahara**

Response: The reviewer comment has been implemented in the manuscript.

**Comment at Line 335. Photo-induced**

Response: The reviewer comment has been implemented in the manuscript.

**Comment at Line 370. Although the peak in PM2.5 could be an outlier, it could also be real reflecting Fe from a different mixture of sources to the days either side, thus illustrating one of the challenges of modelling such a dynamic parameter. I wonder if you could add a few words to the end of line 370 to reflect this.**

Response: We agree with the reviewer. Lines 372-378 have been updated as follows:

On the other hand, the model could not reproduce the peak in total Fe concentration (1.8% of Fe content in $PM_{2.5}$ sample) reported around 29 December 2008. The total Fe observed in $PM_{10}$ (430 ng m$^{-3}$) on 29 December 2008 is lower than that measured on the day before (667 ng m$^{-3}$) and the day after (773 ng m$^{-3}$), whereas that in $PM_{2.5}$ peaked on 29 December 2008 (Srinivas et al., 2012). Thus, the extreme value recorded only for $PM_{2.5}$ on this date may be an outlier. But we do not have sufficient data to confirm this. One of the possibilities is that the sample collected aerosol particles from a mixture of different aerosols sources (e.g., dust and anthropogenic aerosol). This reflects one of the challenges of modelling such a dynamic parameter.